# State & Image Guidance: Teaching Old Text-to-Video Diffusion Models New Tricks

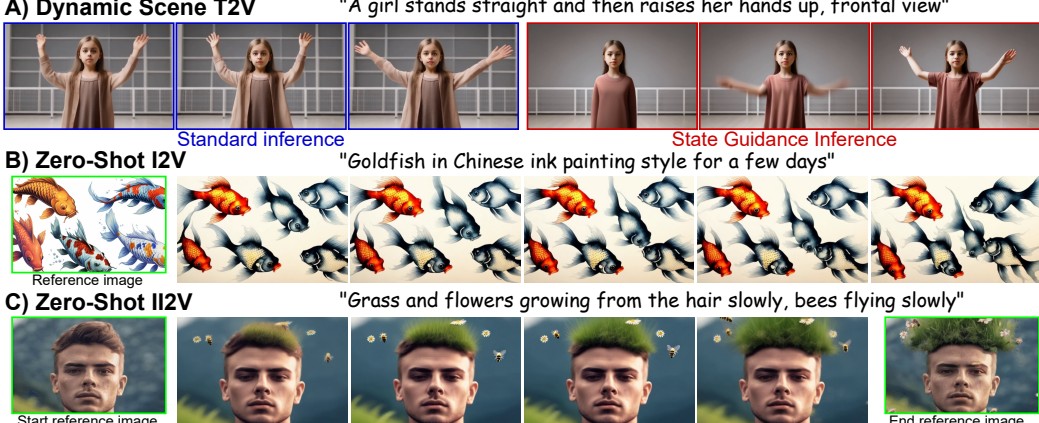

Figure 1: We introduce State Guidance and Image Guidance — two novel sampling methods for T2V diffusion models that enhance their generative capabilities: A) Enable generation of dynamic video scenes; B) Enable zero-shot generation conditioned on the input image (I2V); C) Enable zero-shot generation conditioned on the first and last frames (II2V). The results are generated using VideoCrafter2 Chen et al. (2024a).

## Abstract

Current text-to-video (T2V) models have made significant progress in generating high-quality video. However, these models are limited when it comes to generating dynamic video scenes where the description can vary dramatically from frame to frame. Changing the colour, shape, position and state of objects in the scene is a challenge that current video models cannot handle. In addition, the lack of an inexpensive image-based conditioning mechanism limits their creative application. To address these challenges and extend the applicability of T2V models, we propose two innovative approaches: **State Guidance** and **Image Guidance**. **State Guidance** uses advanced guidance mechanisms to control motion dynamics and scene transformation smoothness by navigating the diffusion process between a state triplet ⟨*initial state, transition state, final state*⟩. This mechanism enables the generation of dynamic video scenes (Dynamic Scene T2V) and allows to control the speed and expressiveness of the scene transformation by introducing temporal dynamics through a guidance weighting schedule over video frames. **Image Guidance** enables Zero-Shot Image-to-Video generation (Zero-Shot I2V) by injecting reference image noise predictions into the initial diffusion steps. Furthermore, the combination of **State Guidance** and **Image Guidance** allows zero-shot transitions between two input reference frames of a video (Zero-Shot II2V). Finally, we introduce the novel **Dynamic Scene Benchmark** to evaluate the ability of the models to generate dynamic video scenes. Extensive experiments show that **State Guidance** and **Image Guidance** successfully address the aforementioned challenges and significantly improve the generation capabilities of existing T2V architectures.

# 1 INTRODUCTION

Text-to-Video (T2V) generation is a rapidly growing area of computer graphics that aims to generate photorealistic videos from input text prompt. These generated videos have tremendous potential to revolutionize video content creation, from personalized short videos to CGI effects and the movie industry.

Despite the rapid advancements in T2V models, significant room for improvement remains. Current T2V generation techniques are primarily limited to synthesizing simple scenes and often lack visual details and dynamic motion Zeng et al. (2023); Qing et al. (2023); Yuan et al. (2024). These models particularly struggle with videos that require distinctly different textual descriptions for the first and last frames, especially in dynamic scenes (see Figure 2). We identify two main reasons for this limitation. First, pre-trained T2V models are rarely trained extensively on dynamic scenes due to their scarcity in training datasets Bain et al. (2021); Chen et al. (2024b). Second, major part of T2V models use the text conditioning mechanism inherited from T2I models Singer et al. (2022); Ho et al. (2022); Blattmann et al. (2023b) that conditions each frame on a uniform text prompt intended to describe the entire video sequence. As a result, frames lack variability and uniqueness. Moreover, standard T2V models do not typically support image conditioning, limiting their general applicability. Implementing image conditioning often requires developing a separate, resource-intensive model without offering a universal solution Xing et al. (2023); Blattmann et al. (2023a).

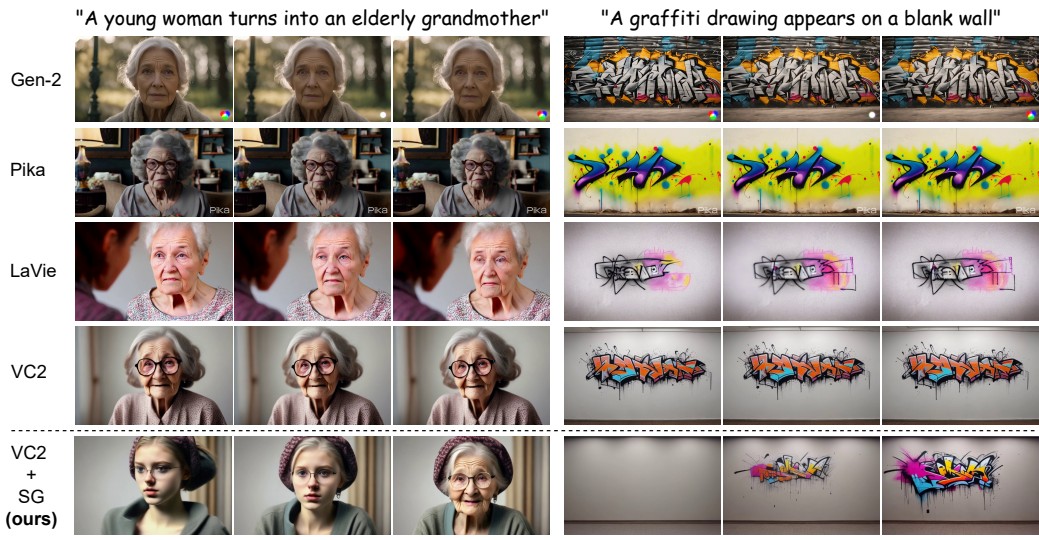

Figure 2: Contemporary T2V models fail to generate videos with scene progression over time ignoring state dynamics described in the text prompt. **State Guidance** enables dynamic scene generation. Models from top to bottom: Gen-2 RunwayML (2024), Pika Pika Labs (2024), LaVie Wang et al. (2023), VideoCrafter2 Chen et al. (2024a), VC2 + SG denote VideoCrafter2 with **State Guidance** inference approach.

To tackle the stated above problems, we propose two novel approaches - **State Guidance** and **Image Guidance** aimed to extend possibilities of stantard T2V models in a training-free manner. Both methods are built upon a modified diffusion sampling process via the guidance mechanism. **State Guidance** provides an alternative view on the T2V model text conditioning. It defines a video scene as a trajectory that has a start point - the *initial state*, an end point - the *final state*, and a trajectory of motion - the *transition state*. As a result, a video scene is described by a state triplet ⟨*initial state, transition state, final state*⟩. During each diffusion step, **State Guidance** makes step in direction of each state simultaneously with different strengths for each frame. Scheduling State Guidance strengths across video frame dimension allows to control video scene dynamic so that first frame corresponds to *initial state*, last frame corresponds to *final state*, and intermediate frames smoothly transition between them. The proposed inference scheme enables T2V models to generate dynamic scenes (see Figure 1A and Figure 2).

**Image Guidance** is a method that enables the injection of image conditions into a pre-trained T2V model without the need for retraining. This process is accomplished by integrating the denoising trajectory with the conditional image in the diffusion trajectory. **Image Guidance** facilitates T2V models to operate in a zero-shot Image-to-Video (I2V) regime (see Figure 1B). Furthermore, the combination of **State Guidance** and **Image Guidance** allows for zero-shot video generation conditioned on the start frame, end frame, and text prompt (Zero-Shot II2V, see Figure 1C).

To evaluate the ability of T2V models to generate dynamic video scenes, we introduce a novel **Dynamic Scene Benchmark**, consisting of 106 textual descriptions for various dynamic scenes. Extensive experiments show that State Guidance significantly improves the generation capabilities of existing T2V architectures, enhancing text understanding and motion quality without a notable decline in temporal consistency. Additionally, we compare our Zero-Shot I2V and II2V regimes with training-based approaches, demonstrating that pre-trained T2V models augmented with State Guidance and Image Guidance can achieve comparable results.

**Contributions:** 1) We introduce **State Guidance**, a novel, training-free framework for T2V diffusion model inference that enables dynamic video scene generation (see Figure 1A and Figure 2); 2) We propose **Image Guidance**, a sampling technique that allows T2V models to condition on images without the need for retraining, facilitating Zero-Shot I2V generation (refer to Figure 1B); 3) We present a zero-shot image-to-video (Zero-Shot II2V) pipeline built upon a pre-trained T2V model, leveraging the combined strengths of **State Guidance** and **Image Guidance** methods (refer to Figure 1C) 4) We present the **Dynamic Scene Benchmark**, the first benchmark in the literature specifically designed to evaluate the ability of T2V models to generate dynamic video scenes.

## 2    RELATED WORK

**Text-to-Video generation.**    Recent breakthroughs in T2I generation using diffusion models have significantly advanced T2V generation. Major T2V models extend T2I architectures by leveraging pre-trained weights and adding temporal layers for frame consistency Blattmann et al. (2023b); Guo et al. (2023); Girdhar et al. (2023); Qing et al. (2023); Zeng et al. (2023). They typically integrate temporal convolutional and attention layers into the 2D UNet of a Stable Diffusion model Rombach et al. (2022), generating in the latent space of a pre-trained VAE Esser et al. (2021); Rombach et al. (2022). While this approach enhances training efficiency and reduces costs, it restricts scene variation by conditioning each frame on a single prompt. Consequently, standard T2V models struggle to produce video scenes with significant frame-by-frame variability. *Our work introduces an innovative sampling mechanism for T2V models, enabling dynamic scene generation in pre-trained models without the need for retraining.*

**Image-to-Video generation.** A natural way to enhance the capabilities and improve the controllability of the T2V model is through the incorporation of image conditioning. This involves extending the architecture and training for this new task Blattmann et al. (2023a); Girdhar et al. (2023); Xing et al. (2023); Zeng et al. (2023); Zhang et al. (2023). For example, I2VGen-XL Zhang et al. (2023) and DynamiCrafter Xing et al. (2023) add cross-attention layers for input image conditioning. EmuVideo Girdhar et al. (2023) and PixelDance Zeng et al. (2023) modify the 3D U-Net by integrating first-frame latent features into the input noise. Stable Video Diffusion Blattmann et al. (2023a) replaces text embeddings with CLIP image embeddings and combines a noisy first frame with the 3D U-Net input. *In contrast to prior works, our sampling method allows a pre-trained T2V diffusion model to perform zero-shot I2V generation without additional optimization or fine-tuning.*

**Image-Image-to-Video generation.** Transient video generation from two images (II2V generation) is a newly explored task in video diffusion models. PixelDance Zeng et al. (2023) trains a model to generate videos using the first and last frames with textual instructions. SIENE Chen et al. (2023) employs a random mask model for text-guided scene transitions, while DiffMorpher Zhang et al. (2024a) uses LoRA parameter interpolation for smooth semantic shifts. TVG Zhang et al. (2024b) builds its II2V pipeline on the pre-trained I2V model DynamiCrafter Xing et al. (2023) model using Gaussian process regression. *In contrast, we demonstrate the feasibility of a zero-shot II2V model on a T2V framework without architectural changes or fine-tuning.*

**Text-to-Video benchmarks.** A conventional T2V evaluation approach assesses the quality of generated frames using FVD Unterthiner et al. (2019) and IS Salimans et al. (2016), while measuring

text similarity with CLIPSIM Radford et al. (2021). However, recent studies indicate that these metrics have a weak correlation with human ratings Girdhar et al. (2023). To address this, several papers propose advanced benchmarks to evaluate generation quality Liu et al. (2024b;a); Huang et al. (2023); Wu et al. (2024). Notably, EvalCrafter Liu et al. (2024a) evaluates videos across four key parameters: visual quality, text-video alignment, motion quality, and temporal consistency. VBench Huang et al. (2023) evaluates using 16 parameters linked to specific prompts, while FETV Liu et al. (2024b) introduces automatic metrics like UMT Score and FVD-UMT, correlating better with user ratings. Despite these advancements, existing benchmarks primarily focus on low-dynamic video scenes, where the description of a single frame applies to the entire video. To address this limitation, *we propose a new benchmark called the **Dynamic Scenes Benchmark**, which emphasizes videos featuring substantial scene progression from frame to frame.*

**Diffusion guidance.** An important feature of Diffusion Models is their ability to customize outputs without the need for retraining. Diffusion Guidance is a technique that modifies the backward diffusion trajectory by adjusting the outputs of the denoising model. Classifier guidance Dhariwal & Nichol (2021) facilitates class-conditional generation from an unconditional model by utilizing gradients from a pre-trained classifier during sampling. Classifier-Free Guidance Ho & Salimans (2021) allows for a balance between sample quality and diversity by combining class-conditional and unconditional estimations and controlling the weight of their mixture. Moreover, MUSE Chang et al. (2023) and MDTv2 Gao et al. (2023) introduce a dynamic guidance scale that changes over the course of the sampling process, resulting in samples with greater diversity in the early steps and higher fidelity in the later stages. *In this work, we propose a guidance schedule across the frame dimension of the generated video to manipulate the dynamics of the video effectively and mixing backward diffusion trajectory with denoising direction to inject image conditioning in the T2V model.*

## 3 METHOD

### 3.1 BACKGROUND

**Diffusion Models.** A diffusion model Ho et al. (2020); Song et al. (2020) is a neural network $\epsilon_\theta$ that is trained to denoise a noisy data $z_t = \sqrt{\alpha_t} z_0 + \sqrt{1 - \alpha_t} \epsilon$ point into $z_0$ a clean data point using a mean squared error loss $\mathcal{L} = \mathbb{E}_{\epsilon,t}[\|\epsilon_\theta(z_t, t, c) - \epsilon\|_2^2]$, where $t$ is diffusion time step, $c$ denotes conditioning, $\epsilon \sim \mathcal{N}(0, \mathbf{I})$ is a noise added to a data point, $\{\alpha_t\}_{t=0}^{T=1}$ denote a noise scales schedule. Trained denoising network $\epsilon_\theta$ enables an Markov chain transitions $q(z_{t-1}|z_t)$ between diffusion time steps called *generative process*, or the *backward diffusion process*. Iterative applying backward diffusion transitions allows to sampling $z_0$ from pure noise $z_T \sim \mathcal{N}(0, \mathbf{I})$. Diffusion model is connected with noise-conditioned score network Song & Ermon (2019) $s_\theta$ that is trained to estimate gradients of the data distribution $s_\theta(z_t, t) \approx \nabla_z \log q(z)$. It can be shown that $\epsilon_\theta(z_t, t) = -\sqrt{1 - \alpha_t} s_\theta(z_t, t)$. Therefore, trained $\epsilon_\theta(z_t, t)$ provides access to a estimation of score function.

**T2V architecture limitation.** T2V models aim to model the conditional data distribution $p(z|p_c)$. This allows for the generation of a coherent video sequence $z = \{z^f\}_{f=1}^F$, where $z^f$ is video frame (or its VAE latent Rombach et al. (2022)) given a conditional text prompt $p_c$.

In this paper, we identify a key limitation in modern T2V models: their inability to generate videos with dynamically changing scenes. Specifically, these models struggle to produce video scenes, where the description of the first frame $z^1$ and the last frame $z^F$ differ significantly. For example, the prompt *"A young woman turns into an elderly grandmother"* should result in the initial frames depicting *"A young woman"* and the final frames showing *"An elderly grandmother"*. However, as illustrated in Figure 2, both commercial models like Gen-2 RunwayML (2024) and Pika Labs (2024), as well as open-source models like LaVie Wang et al. (2023) and VideoCrafter2 Chen et al. (2024a), uniformly misinterpret this prompt, rendering all frames as "An elderly grandmother". We attribute this limitation to two main factors: (1) Training Data: Current T2V models are trained on datasets predominantly composed of static video scenes Bain et al. (2021); Chen et al. (2024b). (2) Model Architecture: Many contemporary T2V models Singer et al. (2022); Ho et al. (2022); Blattmann et al. (2023b) rely heavily on spatial cross-attention between text and latent features for text-guided generation. This method imposes a strong prior, resulting in all frames of the generated video sharing the same description.

In this section, we present a pioneering T2V model inference approach, termed **State Guidance**, which significantly improves the model's capability to generate dynamic video scenes (see Figure 2). Additionally, we introduce **Image Guidance**. When combined with State Guidance, these methods empower the T2V model to achieve zero-shot I2V and zero-shot II2V generation.

## 3.2 STATE GUIDANCE

To address the aforementioned problem, we introduce *State Guidance* - a novel sampling approach for T2V models that requires no architectural modifications or fine-tuning. First, we define a dynamic video scene as a trajectory with an *initial state*, a *transition state*, and a *final state*. The state triplet ⟨*initial state, transition state, final state*⟩ is represented by three prompts $\langle p_{is}, p_{ts}, p_{fs} \rangle$, each corresponding to one state. This triplet can be derived from the original prompt $p_c$ through manual rewriting or automatic generation via LLMs (see Appendix A.4). Second, we adapt the sampling model from $p(z|p_c)$ to $p(z|\langle p_{is}, p_{ts}, p_{fs} \rangle)$, allowing different impacts of $\langle p_{is}, p_{ts}, p_{fs} \rangle$ on each frame, using the score-based formulation of a diffusion model:

$$\nabla_z \log p(z_t|\langle p_{is}, p_{ts}, p_{fs} \rangle) = \nabla_z (\log p(z_t|p_{is}) + \log p(z_t|p_{ts}) + \log p(z_t|p_{fs})) = \\ \nabla_z \log p(z_t|p_{is}) + \nabla_z \log p(z_t|p_{ts}) + \nabla_z \log p(z_t|p_{fs}) \quad (1)$$

The equation above demonstrates that if we have a diffusion model that approximates $\nabla_z \log p(z_t|p_c)$, we can also approximate $\nabla_z \log p(z_t|\langle p_{is}, p_{ts}, p_{fs} \rangle)$. To introduce fine-grained control to either encourage or discourage the model to consider the conditioning information from each element of the state triplet $\langle p_{is}, p_{ts}, p_{fs} \rangle$ in relation to each video frame $f$, we scale each component of the equation by the frame-wise hyperparameters $\gamma_{is}^f, \gamma_{ts}^f, \gamma_{fs}^f$:

$$\nabla \log p^f(z_t|\langle p_{is}, p_{ts}, p_{fs} \rangle) = \gamma_{is}^f \nabla \log p^f(z_t|p_{is}) + \gamma_{ts}^f \nabla \log p^f(z_t|p_{ts}) + \gamma_{fs}^f \nabla \log p^f(z_t|p_{fs}) \quad (2)$$

By reverting to the definition of a diffusion model through the noise prediction network $\epsilon_\theta$ and integrating Equation 2 with Classifier-Free Guidance Ho & Salimans (2021), we arrive at the final formulation of **State Guidance**:

$$\hat{\epsilon}_\theta^f(z_t, \langle p_{is}, p_{ts}, p_{fs} \rangle) = (w + 1) \cdot \tilde{\epsilon}_\theta^f(z_t, \langle p_{is}, p_{ts}, p_{fs} \rangle) - w \cdot \epsilon_\theta^f(z_t, \varnothing) \\ \tilde{\epsilon}_\theta^f(z_t, \langle p_{is}, p_{ts}, p_{fs} \rangle) = \gamma_{is}^f \cdot \epsilon_\theta^f(z_t, p_{is}) + \gamma_{ts}^f \cdot \epsilon_\theta^f(z_t, p_{ts}) + \gamma_{fs}^f \cdot \epsilon_\theta^f(z_t, p_{fs}) \quad (3)$$

Varying the values of $\gamma_{is}^f, \gamma_{ts}^f, \gamma_{fs}^f$ across the dimension of video frames $f$ facilitates smooth transitions between states in the generated dynamic video scenes. In our experiments, we employ Partial Linear and Negative Linear schedules for the prompt triplet $\langle p_{is}, p_{ts}, p_{fs} \rangle$, as detailed in Table 1.

**Guidance interval** We observe that in some cases State Guidance can generate dynamic video scenes with completely unrelated initial and final states (see Figure 3A). We solve this problem by exploiting the fact that diffusion models generate a global scene details at early stages. Thus, we perform first denoising iterations $t \geq \xi$ without State Guidance by conditioning only on $p_{ts}$

Table 1: Guidance schedule description.

| Schedule | Frame index | $\gamma_{is}^f$ | $\gamma_{fs}^f$ | $\gamma_{ts}^f$ |
|---|---|---|---|---|
| Partial linear | $f \in [1, \frac{F}{2}]$ | linear from 1 to 0 | 0 | $1 - \gamma_{is}^f - \gamma_{fs}^f$ |
| | $f \in [\frac{F}{2}, F]$ | 0 | linear from 0 to 1 | |
| Negative linear | $f \in [1, \frac{F}{2}]$ | linear from 2 to 0 | linear from -1 to 0 | $1 - \gamma_{is}^f - \gamma_{fs}^f$ |
| | $f \in [\frac{F}{2}, F]$ | linear from 0 to -1 | linear from 0 to 2 | |
| Partial quadr. | $f \in [1, \frac{F}{2}]$ | quadr. from 1 to 0 | 0 | $1 - \gamma_{is}^f - \gamma_{fs}^f$ |
| | $f \in [\frac{F}{2}, F]$ | 0 | quadr. from 0 to 1 | |

$(\gamma^f_{ts} = 1, \gamma^f_{is} = 0, \gamma^f_{fs} = 0)$. Then, at $t < \xi$, we turn on State Guidance, and the guide video frames to the predefined states. Figure 3B shows that it allows to synchronize initial and final state scenes.

"The big storm stops and the sun comes through the clouds"

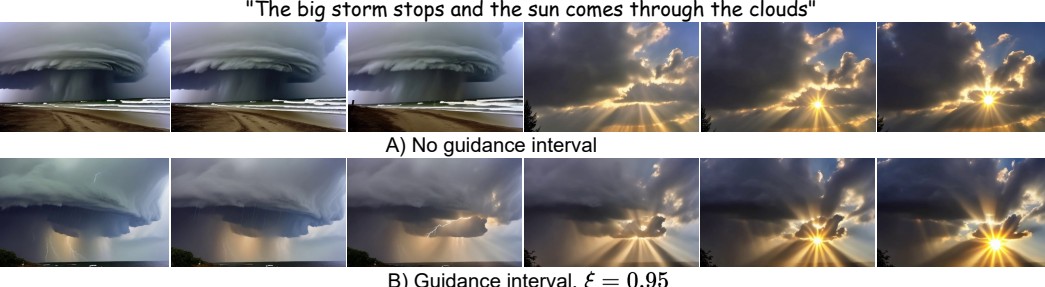

A) No guidance interval

B) Guidance interval, $\xi = 0.95$

Figure 3: **State Guidance** may lead to video scenes that combine two poorly related videos (A). To resolve this issue, we turn off guidance during first steps (B).

### 3.3 IMAGE GUIDANCE

Standard pre-trained T2V do not support image conditions. To address this issue, we introduce **Image Guidance** that injects image conditions $i_c$ into pre-trained T2V model convert sampling model from $p(z|p_c)$ to $p(z|p_c, i_c)$. To do so, similar to Section 3.2 we use a score-based formulation of a diffusion model:

$$\nabla \log p(z|p_c, i_c) = \nabla(\log p(z_t|p_c) + \log p(z_t|i_c)) = \eta \nabla \log p(z_t|p_c) + (1-\eta)\frac{\sqrt{\alpha_t} \cdot i_c - z^f_t}{1 - \alpha_t} \quad (4)$$

Where $\eta$ is a parameter that controls image guidance strength that we set equal to 0.7. By reverting to the definition of a diffusion model through the noise prediction network $\epsilon_\theta$ **Image Guidance** takes form:

$$\bar{\epsilon}^f(z^f_t, p_c, i_c) = \eta \cdot \epsilon_\theta(z_t, p_c) + (1-\eta) \cdot \bar{\epsilon}^f(z^f_t, i_c) = \eta \cdot \epsilon_\theta(z_t, p_c) + (1-\eta)\frac{z^f_t - \sqrt{\alpha_t} \cdot i_c}{\sqrt{1 - \alpha_t}} \quad (5)$$

Consequently, by mixing a denoising direction that from $z^f_t$ to $i_c$ with denoising network prediction $\epsilon_\theta(z_t)$, we can induce image conditioning into backward diffusion process.

**Zero-Shot I2V Generation.** To facilitate Zero-Shot I2V Generation using T2V models, we modify the backward diffusion process as follows:

$$\tilde{\epsilon}^f_\theta(z_t, p_c, i_c) = \begin{cases} \bar{\epsilon}^f(z^f_t, \varnothing, i_c), & t \geq \xi \\ \epsilon^f_\theta(z_t, p_c), & t < \xi \end{cases} \quad (6)$$

During first diffusion timesteps $t \geq \xi$, we form a scene layout that is semantically close to a reference image $i_c$ using **Image Guidance**. During further diffusion timesteps $t < \xi$, we generate a temporal dynamics defined in prompt $p_c$ using standard T2V model sampling. We also combine Equation 6 with Classifier-Free Guidance Ho & Salimans (2021).

### 3.4 COMBINING STATE AND IMAGE GUIDANCE

**State Guidance** (Equation 3) allows to transform a sampling $p(z|p_c)$ into $p(z|\langle p_{is}, p_{ts}, p_{fs}\rangle)$. Adding **Image Guidance** (Equation 5) to this combination allows to obtain $p(z|\langle i_{is}, p_{ts}, i_{fs}\rangle)$ sampling model, where $i_{is}$ and $i_{fs}$ are reference image for the first and the last frame on the video. In other words, applying combination of **State Guidance** and **Image Guidance** to pre-trained T2V model, enables **Zero-Shot Image-Image-to-Video (II2V) Generation**:

$$\hat{\epsilon}_\theta^f(z_t, \langle i_{is}, p_{ts}, i_{fs} \rangle) = (w+1) \cdot \tilde{\epsilon}_\theta^f(z_t, \langle i_{is}, p_{ts}, i_{fs} \rangle) - w \cdot \epsilon_\theta^f(z_t, \varnothing)$$

$$\tilde{\epsilon}_\theta^f(z_t, \langle i_{is}, p_{ts}, i_{fs} \rangle) = \gamma_{is}^f \cdot \bar{\epsilon}_\theta^f(z_t, p_{ts}, i_{is}) + \gamma_{ts}^f \cdot \epsilon_\theta^f(z_t, p_{ts}) + \gamma_{fs}^f \cdot \bar{\epsilon}_\theta^f(z_t, p_{ts}, i_{fs}) \tag{7}$$

**Guidance strength schedule.** We use quadratic guidance schedule (described in Table 1). We also use guidance interval to achieve a better temporal consistency of the scene, during the first $t \geq \xi$ diffusion time steps, we set $\gamma_{is}^f = \frac{f}{F}, \gamma_{fs}^f = 1 - \frac{f}{F}, \gamma_{ts}^f = 0$.

### 3.5 DYNAMIC SCENES BENCHMARK

Our literature review has shown that there is a lack of benchmarks to evaluate a dynamic scene generation for T2V models. To fill this gap, we present **Dynamic Scenes Benchmark** - a collection of prompts for dynamic scene generation. We manually collect 106 prompts that describe a video scene with noticeable scene changes throughout the video. We divide scene changes into two broad categories: object property changes and object position changes. Object property changes include a wide variety of possible scenarios: object growth time lapse (plant, animal, human), color change, human mood change, weather change, etc. Object position change has only two types of changes: object position change, objects appear/disappear. For each prompt in prompt list we provide Initial state, Transition state, and End state text descriptions, the example is shown in Table 2.

Table 2: Example of samples in **Dynamic Scenes Benchmark**.

| Prompt | Initial state | Transition state | End state |
|---|---|---|---|
| *Empty glass fills with water* | *An empty glass* | *A glass is being filling with water* | *A glass with water* |
| *The foggy forest landscape, the fog lifts and it's clear and sunny* | *The foggy forest landscape* | *The forest landscape, the fog is lifting and it's clear and sunny* | *The clear and sunny forest landscape* |
| *A girl stands straight and then raises her hands up, frontal view* | *A girl stands straight, frontal view* | *A standing girl is raising her hands up, frontal view* | *A girl stands with hands raised up, frontal view* |

## 4 EXPERIMENTS

**Implementation details:** We evaluate possibilities of State Guidance and Image Guidance, by combining them with three representative open-source T2V models: VideoCrafter2 Chen et al. (2024a) and base LaVie Wang et al. (2023), that generate 16-frame videos in $320 \times 512$ resolution and CogVideoX-5B Yang et al. (2024) generate 49-frame videos in $480 \times 720$. Code and checkpoints are taken from their official GitHub repositories: generation team of Shanghai AI Laboratory. Partner with OpenGVLab (2024), Center (2024). VideoCrafter2 Chen et al. (2024a), base LaVie Wang et al. (2023), and CogVideoX-5B at Tsinghua University. (2024) were inferenced with 50 steps DDIM sampling Song et al. (2020), other models were inferenced with their default parameters. All generations were performed locally on a single Nvidia A100 80 Gb GPU with frozen random state or using the available generative models API.

**Metrics:** We quantitatively evaluate generated videos by estimating: Text Similarity – TextSim, average absolute Optical Flow – OF Score, Temporal consistency – TC, and Image Similarity – ImSim (used for I2V experiments). We estimate TextSim using UMT Score Liu et al. (2024b). This metric uses Vision-Language Model (VLM) Li et al. (2024) and shows superior correlation with human evaluations Liu et al. (2024b). OF Score estimates amount of motion in the video and is calculated by averaging absolute value of optical flow map predicted by RAFT large model Teed & Deng (2020). TC is calculated by averaging CLIP Radford et al. (2021) similarity between the subsequent frames of the video. ImSim is calculated by averaging CLIP similarity between the generated video frames and reference image.

**Benchmarks:** 1) We analyze T2V dynamic video generation capabilities via the *Dynamic Scenes Benchmark* defined in Section 3.5, measuring TextSim, OF Score, and TC. 2) For the I2V evaluation,

Table 3: Dynamic scene T2V generation quantitative results. SG columns indicates whether State Guidance inference scheme was used or not. For all models with State Guidance we user Negative linear guidance schedule and $\xi = 0.95$.

| Model | SG | TextSim ↑ | OF Score ↑ | TC, % ↑ |
|---|---|---|---|---|
| Gen-2 RunwayML (2024) | ✗ | 2.64 | 1.23 | 99.3 |
| Pika Labs (2024) | ✗ | 2.58 | 1.76 | 98.9 |
| FreeBloom Huang et al. (2024) | ✗ | 2.63 | 3.40 | 92.3 |
| DirecT2V Hong et al. (2023) | ✗ | 2.50 | 49.41 | 86.8 |
| LaVie Wang et al. (2023) | ✗ | 2.80 | 4.78 | **98.2** |
| | ✓ | **3.10** | **9.24** | 96.8 |
| VideoCrafter2 Chen et al. (2024a) | ✗ | 2.87 | 2.07 | **98.4** |
| | ✓ | **3.18** | **3.83** | 97.4 |
| CogVideoX Yang et al. (2024) | ✗ | 2.85 | **3.10** | 98.3 |
| | ✓ | **3.01** | 1.72 | 98.0 |
| CogVideoX (PE) Yang et al. (2024) | ✗ | 3.01 | **3.19** | **98.71** |
| | ✓ | **3.16** | 2.19 | 98.32 |

we manually collected a *Custom I2V Benchmark* comprising 111 image-prompt pairs from five open-domain I2V methods (Girdhar et al. (2023) - 4, Gong et al. (2024) - 20, Xing et al. (2023) - 22, Zeng et al. (2023) - 24, Zhang et al. (2023) - 41). The metrics assessed include TextSim, ImSim, OF Score, and TC. 3) II2V evaluations were executed using *MorphBench* Zhang et al. (2024a), where we assessed the fidelity and smoothness of the video output using traditional metrics such as Frechet Inception Distance (FID) Heusel et al. (2017) and Perceptual Path Length (PPL) Karras et al. (2020), further details of which can be found in the Appendix A.5.

## 4.1 Dynamic scene T2V generation

We compare VideoCrafter2 Chen et al. (2024a), LaVie Wang et al. (2023), CogVideoX Yang et al. (2024), and CogVideoX (PE) Yang et al. (2024) on the Dynamic Scenes Benchmark both under standard inference and with State Guidance. CogVideoX (PE) enhances both the original prompt and state prompt triplets using the CogVideoX prompt enhancer from CogVideoX-5B-Space. Quantitative results (Table 3) and user studies (Table 4) show that State Guidance improves the alignment between generated videos and prompts and enhances video dynamism, with only a negligible decrease in temporal consistency (TC). This minor reduction is expected, as the TC metric favors static videos. Qualitative effects of State Guidance are illustrated in Figure 1A and Figure 2 in the Supplementary materials. Details on the user study and analysis of State Guidance hyperparameters are provided in Appendix A.3. Qualitatively, the effect of State Guidance can be seen in the Figure 1A and Figure 2.

Additionally, we include reference results for two commercial T2V frameworks: Gen-2 RunwayML (2024) and Pika Labs (2024), as well as two models that utilize multiple prompts generated by LLM to enhance video generation: FreeBloom Huang et al. (2024) and DirecT2V Hong et al. (2023). Table 3 shows that all these methods exhibit low TextSim, indicating their failure to correctly generate dynamic video scenes (see Figure 2). While Gen-2 and Pika demonstrate higher TC scores, this can be attributed to their tendency to produce videos with reduced dynamics, as evidenced by low OF Scores. In contrast, DirecT2V achieves the highest OF Score, though this is accompanied by inconsistencies in video output (with a TC score below 87).

Table 4: Dynamic scene T2V generation user study results. ✓SG: percentage preferring State Guidance inference; ✗SG: percentage preferring standard inference; Equal: percentage rating both equally. For all models with State Guidance we user Negative linear guidance schedule and $\xi = 0.95$.

| Model | Text Alignment | | | Dynamism | | |
|---|---|---|---|---|---|---|
| | ✓SG , % | Equal, % | ✗SG, % | ✓SG, % | Equal, % | ✗SG, % |
| LaVie Wang et al. (2023) | **70.6** | 13.0 | 16.4 | **74.3** | 5.1 | 20.6 |
| VideoCrafter2 Chen et al. (2024a) | **66.7** | 22.2 | 11.1 | **68.1** | 14.3 | 17.6 |
| CogVideoX Yang et al. (2024) | **42.8** | 22.6 | 34.6 | 41.0 | 11.1 | **47.9** |
| CogVideoX (PE) Yang et al. (2024) | **57.2** | 17.9 | 24.9 | **50.2** | 9.2 | 40.6 |

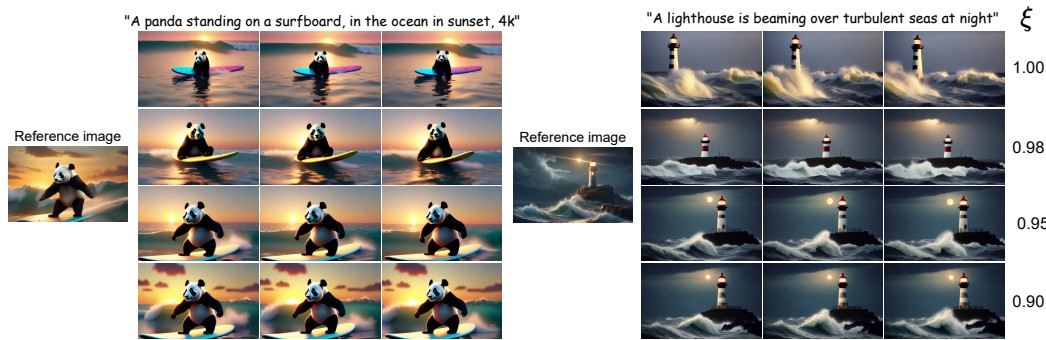

Figure 4: Illustration of zero-shot I2V outputs with VideoCrafter2 + State Guidance with different $\xi$ parameters. Decreasing $\xi$ increases image similarity and decreases motion.

## 4.2 ZERO-SHOT I2V

We evaluate our Zero-Shot Image-to-Video (I2V) pipeline, which is built upon the pre-trained T2V model VideoCrafter2 Chen et al. (2024a) using Image Guidance. Table 5 presents a quantitative comparison with another Zero-Shot I2V method, TI2V-Zero Ni et al. (2024) on *Custom I2V Benchmark* described in the beginning of this section. We provide results using Image Guidance with three hyperparameters, $\xi$: 0.98, 0.95, and 0.90. Our findings indicate that our method outperforms TI2V-Zero in terms of Text Similarity, Image Similarity, and Temporal Consistency (TC). Although TI2V-Zero achieves a higher Optical Flow (OF) Score, this comes at the cost of lower temporal consistency, as evidenced by a TC score below 93.

**Hyperparameters analysis:** By varying $\xi$, we can modulate both Image Similarity (ImSim) and Text Similarity (TextSim). Setting $\xi = 1.0$ allows for generation without image conditioning (T2V), while decreasing $\xi$ enhances image similarity and reduces motion in the video (as reflected in the OF Score). This effect is illustrated in Figure 4 and Table 5.

**Comparison with I2V:** We also include in Table 5 results for I2V models that were specially trained for this task: I2VGen-XL Zhang et al. (2023), SVD Zhang et al. (2023), and DynamiCrafter Xing et al. (2023). Our zero-shot pipeline demonstrates superior text similarity and comparable temporal consistency when compared to these training-based methods. However, it shows lower performance in image similarity and OF Score, which is expected for a zero-shot approach.

Table 5: Quantitative Evaluation of I2V Generation on the Custom I2V Benchmark. In the Mode column, I2V and I2V-Z represent training-based and zero-shot image-to-video generation, respectively.

| Model | Mode | $\xi$ | TextSim ↑ | ImSim ↑ | OF Score ↑ | TC, % ↑ |
|---|---|---|---|---|---|---|
| I2VGen-XL Zhang et al. (2023) | | – | 2.99 | 0.919 | 1.86 | 98.9 |
| SVD Zhang et al. (2023) | I2V | – | 2.66 | 0.906 | 4.60 | 97.9 |
| DynamiCrafter Xing et al. (2023) | | – | 2.87 | 0.934 | 1.95 | 99.1 |
| TI2V-Zero Ni et al. (2024) | | – | 3.39 | 0.764 | **20.48** | 92.4 |
| | I2V-Z | 0.98 | **3.49** | 0.799 | 1.17 | 98.8 |
| VC2 + IG (Ours) | | 0.95 | 3.45 | 0.817 | 0.58 | 99.1 |
| | | 0.90 | 3.37 | **0.831** | 0.32 | **99.3** |

## 4.3 ZERO-SHOT II2V

We evaluate our Zero-Shot Image-Image-to-Video (II2V) pipeline, which is built upon the pre-trained T2V model VideoCrafter2 Chen et al. (2024a). This pipeline uses the combination of State Guidance and Image Guidance with $\xi = 0.5$ and a partial quadratic guidance schedule. Table 6 presents a quantitative comparison of our zero-shot pipeline against other II2V models on the *MorphBench* benchmark Zhang et al. (2024a). This comparison includes training-based models such as DynamiCrafter Xing et al. (2023), DiffMorpher Zhang et al. (2024a), and TVG Zhang et al. (2024b), which rely on a pre-trained I2V model in a zero-shot context. Notably, our model operates

without the need for training-based image conditioning, yet achieves robust quantitative results that surpass previous approaches. Figure 5 showcases comparative examples of the generated results. The analysis of hyperparameter $\xi$ selection can be found in Appendix A.5.

Table 6: Quantitative evaluation of II2V generation on MorphBench. We report FID ($\downarrow$) and PPL ($\downarrow$) to assess the fidelity and smoothness of the transition videos, respectively, across the Metamorphosis, Animation, and Overall categories.

| Model | Metamorphosis | | Animation | | Overall | |
|---|---|---|---|---|---|---|
| | FID $\downarrow$ | PPL $\downarrow$ | FID $\downarrow$ | PPL $\downarrow$ | FID $\downarrow$ | PPL $\downarrow$ |
| DynamiCrafter Xing et al. (2023) | 87.32 | 42.09 | 43.31 | 11.16 | 69.13 | 33.84 |
| SEINE Chen et al. (2023) | 82.03 | 47.72 | 48.25 | 16.26 | 67.60 | 39.33 |
| DiffMorpher Zhang et al. (2024a) | 70.49 | 18.19 | 43.15 | **5.14** | 54.69 | 21.10 |
| TVG Zhang et al. (2024b) | 86.92 | 35.18 | 42.99 | 12.46 | 64.05 | 29.08 |
| S&IG (Ours) | **35.46** | **12.26** | **31.44** | 6.58 | **30.15** | **10.75** |

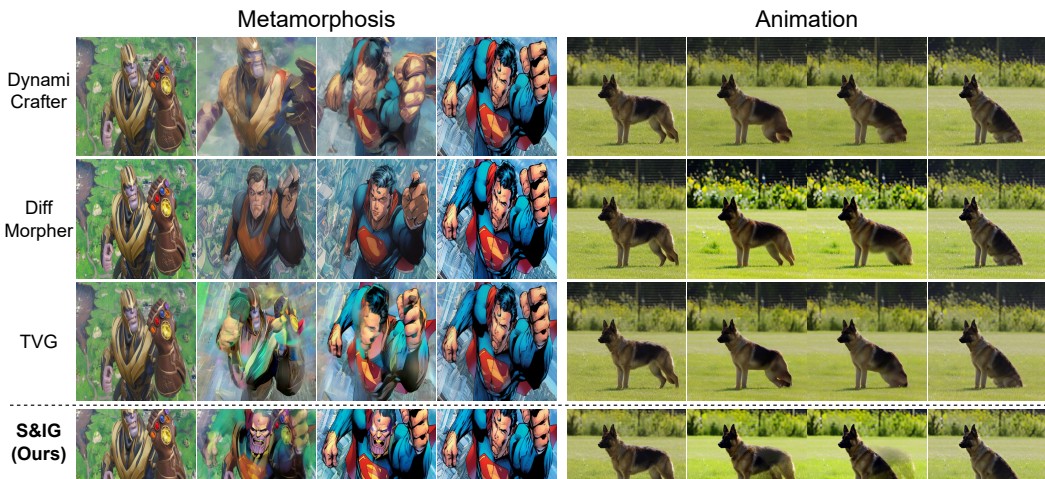

Figure 5: Examples of II2V generations from the *MorphBench* benchmark for Metamorphosis and Animation categories. In contrast to other models, our pipeline employs a method that does not require training-based image conditioning, yet it achieves comparable quality.

## 5 CONCLUSION

In this paper, we introduced two novel sampling methods for T2V diffusion models: **State Guidance** and **Image Guidance**. These methods enhance the capabilities of pre-trained T2V models without requiring additional training or architectural modifications. **State Guidance** enables T2V models to generate dynamic video scenes, overcoming the limitations imposed by their text conditioning mechanisms. The efficiency of the proposed solution has been measured on the proposed first in the literature **Dynamic Scenes Benchmark**. Meanwhile, **Image Guidance** incorporates image conditioning into pre-trained T2V models, allowing them to generate content in a Zero-Shot I2V mode. The combination of **State Guidance** and **Image Guidance** facilitates the generation of zero-shot transition videos based on two reference images and a text prompt, namely Zero-Shot II2V.

While our approach has yielded significant results, there is substantial potential for further research. First, we believe the text conditioning mechanism currently employed in most T2V models has critical shortcomings and should be replaced with more modern architectural techniques. Second, the framework introduced in **State Guidance** can be combined with trainable adapters for state conditioning, which may enhance output video quality and controllability. Finally, the proposed zero-shot II2V and zero-shot I2V schemes can be integrated with existing training-based methods to further improve final video quality.

## 6 ETHICS & REPRODUCIBILITY

The use of T2V foundation models raises several ethical concerns. These models have the potential for misuse, such as generating misleading or counterfeit content, which could have harmful societal impacts. Our work relies heavily on two models, VideoCrafter2 Chen et al. (2024a) and LaVie Wang et al. (2023), making it vulnerable to these risks. Furthermore, the video datasets used to train these models may contain inappropriate content or biases that the models could inadvertently perpetuate, resulting in the generation of inappropriate material. In addition, our *Custom I2V Benchmark* scoring is based on qualitative results from prior work, which could also be misused. To address these concerns and promote reproducibility, we will release our source code and benchmarks under a license that encourages ethical and legal use. Additional information about implementation details, metrics can be found in the Experiments section and in the Appendix.

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

# A  APPENDIX

## A.1  LIMITATIONS

While State Guidance and Image Guidance enhance the pre-trained T2V models by introducing new features and capabilities without the need for retraining, their generation quality is ultimately constrained by the original T2V model. This limitation is illustrated in Figure 6. In the first example, we attempt to generate a video with camera control. However, due to the inherent limitations of standard T2V models in this area Hu et al. (2024), State Guidance inference simply inherits this issue: instead of producing a video with a rotating camera, it results in a video featuring a rotating horse. The second and third examples highlight challenges that the original model struggles to address, such as "the sunflower turning into an astronaut" and "the lorry transforming into a robot." Although State Guidance generates coherent and temporally consistent videos, it often fails to achieve the transformations exactly as requested. We attribute this to a possible lack of relevant transformations in the original training samples of the T2V model.

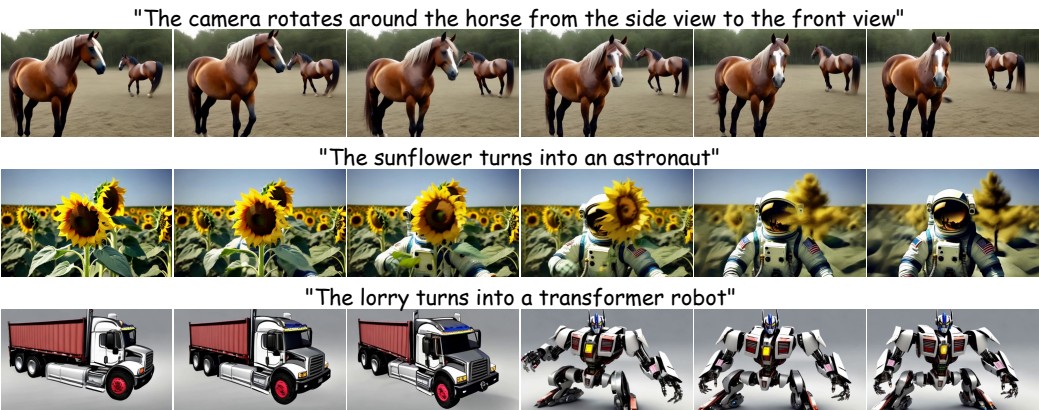

Figure 6: The generation ability of State Guidance is limited by the pre-trained T2V model. *Rows*: (i) *Single prompt*: "the camera rotates around the horse from the side view to the front view", *Prompt triplet*: ⟨ "horse, side view", "the camera rotates around the horse", "horse, front view"⟩; (ii) *Single prompt*: "the sunflower turns into an astronaut", *Prompt triplet*: ⟨ "the sunflower", "the sunflower is turning into an astronaut", "an astronaut" ⟩; (iii) *Single prompt*: "the lorry turns into a transformer robot", *Prompt triplet*: ⟨ "the lorry", "the lorry is turning into a transformer robot", "a transformer robot"⟩. The provided results are generated using VideoCrafter2 and State Guidance.

Another significant limitation is that State Guidance and its combination with Image Guidance requires more model inferences during sampling. While classifier-free guidance demands only two model inferences per diffusion step — one conditional and one unconditional — State Guidance for dynamic video scene generation and combination of State Guidance and Image Guidance for Zero-Shot II2V pipeline require four: three for each state and one unconditional for classifier-free guidance. This increases the inference time and resource consumption. Lastly, while State Guidance and Image Guidance add new features and capabilities to the pre-trained T2V models, they also introduce additional hyperparameters, such as the guidance schedule and guidance interval, complicating the use of T2V models.

## A.2  STATE DYNAMICS ANALYSIS

We demonstrate the impact of State Guidance on scene transitions. To achieve this, we calculate the CLIP similarity between each frame of the video in Figure 7, the original prompt, and each prompt in the state triplet. As shown in Figure 7, videos generated with standard inference exhibit nearly constant CLIP similarity across all frames, indicating a lack of state dynamics. In contrast, videos generated with State Guidance display significant scene progression: the similarity to the first state decreases throughout the video, while the similarity to the last state increases.

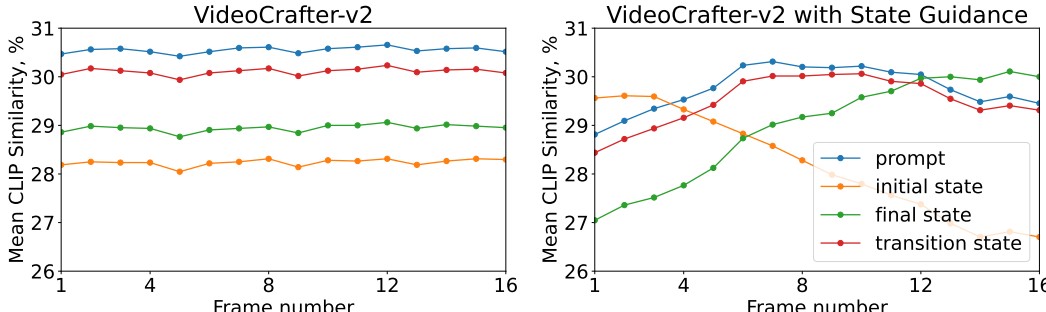

Figure 7: CLIP similarities between prompt, triplet ⟨*initial state, final state, transition state*⟩ for original VideoCrafter2 and VideoCrafter2 with State Guidance (ours). VideoCrafter2 shows nearly the same text alignments to both prompt and triplet states. However, State Guidance injection shows gradually increasing/decreasing of final state/initial state text alignment with frame number.

## A.3 Additional Dynamic scene T2V generation results

**Hyperparameters analysis.** State Guidance introduces additional hyperparameters to T2V model inference: guidance strength schedule ($\{\gamma_{is}^f\}_{f=1}^F, \{\gamma_{ts}^f\}_{f=1}^F, \{\gamma_{fs}^f\}_{f=1}^F$, values across video dimension $f$) and guidance interval parameter $\xi$. Table 7 shows that higher $\xi$ (smaller interval without guidance) leads to higher text similarity and amount of motion in the video. However, inference without guidance interval ($\xi = 0$) may lead to completely unrelated initial and final video scenes (Figure 3). That is why we set $\xi = 0.95$ in dynamic T2V scenes generation. Table 7 also show that using *Negative linear* guidance schedule is strictly better compared to *Partial linear* guidance schedule.

Table 7: Quantitative results for different guidance schedule and guidance interval parameters $\xi$.

| Guidance schedule | $\xi$ | TextSim ↑ | OF Score ↑ | TC, % ↑ |
|---|---|---|---|---|
| Negative linear | 1.00 | 3.19 | 6.00 | 97.1 |
| | 0.95 | 3.18 | 3.83 | 97.4 |
| | 0.90 | 3.12 | 3.31 | 97.4 |
| | 0.80 | 2.97 | 2.07 | 98.0 |
| Partial linear | 1.00 | 2.91 | 2.92 | 98.1 |
| | 0.95 | 2.87 | 2.59 | 98.1 |
| | 0.90 | 2.89 | 2.35 | 98.1 |
| | 0.80 | 2.81 | 1.96 | 98.3 |

**User study details.** Users were asked two questions: "Which video better reflects the actions described in the text description?" and "Which video is more dynamic (has more action and events, including simultaneous events)?". Each question has three options: Video 1, Video 2, or Equal (to account for instances where users are unable to prefer one option over the other). For each side-by-side comparison, between 50 and 67 users participated, with each pair of videos assessed by at least 5 unique users.

**Quantitative results robustness.** To demonstrate the statistical robustness of our results in Table 3, we re-evaluated the metrics in Table 3 for LaVie and VideoCrafter2, with and without State Guidance,

using five different random seeds (see Table 8). The low standard deviations observed affirm robustness, and the non-overlapping value intervals further confirm the consistency of our findings.

Table 8: Dynamic scene T2V generation quantitative results robustness illustration. SG columns indicates whether State Guidance inference scheme was used or not. For all models with State Guidance we user Negative linear guidance schedule and $\xi = 0.95$.

| Model | SG | TextSim ↑ | OF Score ↑ | TC, % ↑ |
|---|---|---|---|---|
| LaVie Wang et al. (2023) | ✗ | 2.77 ± 0.04 | 4.77 ± 0.52 | **97.90 ± 0.20** |
| | ✓ | **3.14 ± 0.05** | **9.00 ± 0.72** | 96.40 ± 0.10 |
| VideoCrafter2 Chen et al. (2024a) | ✗ | 2.84 ± 0.09 | 1.97 ± 0.10 | **98.42 ± 0.04** |
| | ✓ | **3.12 ± 0.05** | **3.70 ± 0.25** | 97.30 ± 0.20 |

### A.4 STATE TRIPLETS GENERATION

In this section, we outline the process of state triplet generation. This can be accomplished either manually by the user adjusting the prompt or automatically using a large language model (LLM). Table 9 presents a quantitative comparison of VideoCrafter2 with standard inference, and VideoCrafter2 with State Guidance sampling with both manually generated state prompts and those generated by the GPT-4o Achiam et al. (2023) model. Additionally, we detail the manual procedures for generating state prompts and provide instructions for automatic generation using GPT-4o.

Table 9: In addition to the results from Table 3 in the main paper, we provide results for State Guidance with state triplets automatically generated by GPT4o from original prompt and report results for FreeBloom and DirecT2V - models that generate prompt for each frame with LLM. It is important to note that State Guidance achives highest TextSim. Large OF Score for DirecT2V is a result of low temporal consistency.

| Model | TextSim ↑ | OF Score ↑ | TC ↑ |
|---|---|---|---|
| VideoCrafter2 | 2.87 | 2.07 | 98.4 |
| VideoCrafter2 + SG (manual prompts) | 3.18 | 3.83 | 97.4 |
| VideoCrafter2 + SG (GPT4o prompts) | 3.01 | 5.15 | 97.2 |

**Manual triplet generation.** We describe the process of manually selecting prompts for our *Dynamic Scene Benchmark* that describe dynamic changes in video scenes. The primary goal is to capture evolving actions or transitions, such as objects changing properties (e.g., flowers blooming, ice melting, or color changing) and changes in position (e.g., a person standing up or a bird flying away). Prompts fall into two categories: those with an active main object that undergoes a clear evolution while the background remains relatively static, and those where the background itself changes without a main object. The key criterion for selection is that the changes must be gradual, allowing for intermediate states, as opposed to instantaneous transitions that would not provide a smooth evolution of motion. This distinction ensures that we focus on motion that can be meaningfully visualized over time.

**Automated triplet generation.** While the triplet conditions for our experiments were generated manually to ensure accuracy, we recognize the importance of automation for reproducibility. We have explored the use of large language models (LLMs) to automate the generation of triplets, specifically with GPT-4o. We begin with serial prompting using the following startup instructions:

Then we use the following prompt to rewrite and consistent linguistic structure in the generated triplets:

### A.5 ADDITIONAL II2V RESULTS

**Metric details:** To quantitatively assess the quality of intermediate images and the smoothness of transition video, we use the metrics adopted in TVG Zhang et al. (2024b) and incorporate some of their results.

**Instruction:**

Given a text prompt for dynamic video scenes, you must create 3 succinct text descriptions that describe that text prompt. Before you write each description, you must follow these instructions. These are of primary importance:

1. Describe an action or event in a dynamic sequence, providing a clear starting state ("initial prompt"), a transition state ("transition prompt") and a contrasting ending state ("final prompt").

2. Use language that implies transformation, evolution or change over time.

3. Linguistic structure of each sentence should be simple and similar.

4. Please be straightforward and do not use a narrative style.

Use the following output format: ["initial prompt", "transition prompt", "final prompt"]

**In-context example:**

**Input:**
**1.** "Empty glass fills with water";
**2.** "A sad woman becomes happy, close-up";
**3.** "Flower blooms from bud to flower";
**4.** "A graffiti drawing appears on a blank wall";

**Output:**
**1.** ["an empty glass", "a glass is being filling with water", "a glass with water"];
**2.** ["a sad woman, close-up", "A sad woman is becoming happy, close-up", "a happy woman, close-up"];
**3.** ["a bud", "a flower is blooming", "a flower"];
**4.** ["a blank wall", "a graffiti drawing appears on a blank wall", "a wall with a graffiti drawing"]

**Input texts:** [*insert list of text prompts here*]

**Instruction:**

Now perform Coreference Resolution on the sentences generated above, replace reflexive pronouns with their original vocabulary, and eliminate the discourse cohesion. Keep the meaning the same. Use the same output format.

- Frechet Inception Distance (FID, ↓) Heusel et al. (2017). The FID score is calculated by comparing the distribution of the input images with that of the generated images. To estimate the generated image distribution, we randomly select two images from the interpolation video 10 times and calculate the average FID score. This serves as an indicator of the accuracy and realism of the intermediate images.

- Perceptual Path Length (PPL, ↓) Karras et al. (2020). We measure the sum of the perceptual loss Zhang et al. (2018) between consecutive frames in the video. This metric reflects the smoothness and consistency of transitions throughout the video.

- Temporal consistency (TC, % ↑) evaluates whether the generated video frames remain coherent and consistent with each other. To measure this, we calculate the CLIP Radford et al. (2021) image similarity between each pair of adjacent frames in the generated video and take the average.

**Hyperparameters analysis:** To validate the effectiveness of our Zero-Shot II2V method, we performed an ablation study with different guidance interval parameters, with the results presented in Table 10. The results show that as the interval parameter increases, the Perceptual Path Length (PPL) also increases, reaching its peak when both the first and last frames influence the predicted noise at each diffusion step ($\xi = 1.0$). This occurs because the transition between the first and last frames in the generated video becomes abrupt, causing most frames to closely resemble either the starting or ending frame. Meanwhile, the FID and Temporal Consistency (TC) metrics stabilize

at an intermediate interval ($\xi = 0.5$), which allows the model to generate a more diverse range of intermediate frames while maintaining a smooth transition between the first and last frames. We select this interval value as our primary parameter. Notably, the metrics exhibit similar trends for the entire DiffBench dataset (overall) and its individual categories (Animation, Metamorphosis).

Table 10: Ablation study. Quantitative results for different guidance interval parameters $\xi$, in the Zero-Shot II2V pipeline S&IG on MorphBench.

| $\xi$ | Metamorphosis | | | Animation | | | Overall | | |
|---|---|---|---|---|---|---|---|---|---|
| | FID $\downarrow$ | PPL $\downarrow$ | TC, % $\uparrow$ | FID $\downarrow$ | PPL $\downarrow$ | TC, % $\uparrow$ | FID $\downarrow$ | PPL $\downarrow$ | TC, % $\uparrow$ |
| 0.0 | 52.30 | 30.77 | 94.02 | 63.89 | 22.60 | 95.65 | 48.43 | 28.59 | 94.45 |
| 0.1 | 43.95 | 22.57 | 95.70 | 43.82 | 11.84 | 97.72 | 38.55 | 19.71 | 96.23 |
| 0.3 | 37.25 | 15.80 | 96.94 | 36.52 | 8.64 | 98.33 | 32.92 | 13.89 | 97.31 |
| 0.5 | **35.46** | 12.26 | **97.21** | **31.44** | 6.58 | 98.61 | **30.15** | 10.75 | **97.58** |
| 0.7 | 36.14 | 10.46 | 97.08 | 33.34 | 5.56 | **98.77** | 32.05 | 9.16 | 97.53 |
| 0.9 | 41.20 | 9.89 | 96.63 | 33.71 | 4.73 | 98.69 | 35.61 | 8.51 | 97.18 |
| 1.0 | 60.58 | **8.27** | 96.80 | 45.14 | **4.00** | 98.76 | 51.33 | **7.13** | 97.32 |

**Comparison with PixelDance**: In Figure 8, we show qualitative results of video generation results conditioned on the first and last video frames. The combination of VideoCrafter2 and State Guidance allows to achieve visual effects comparable to PixelDance Zeng et al. (2023) trained on image-text-image triplets. Unfortunately, code and weights of PixelDance Zeng et al. (2023) are not available, that is why we compare with the generation samples from their project page.

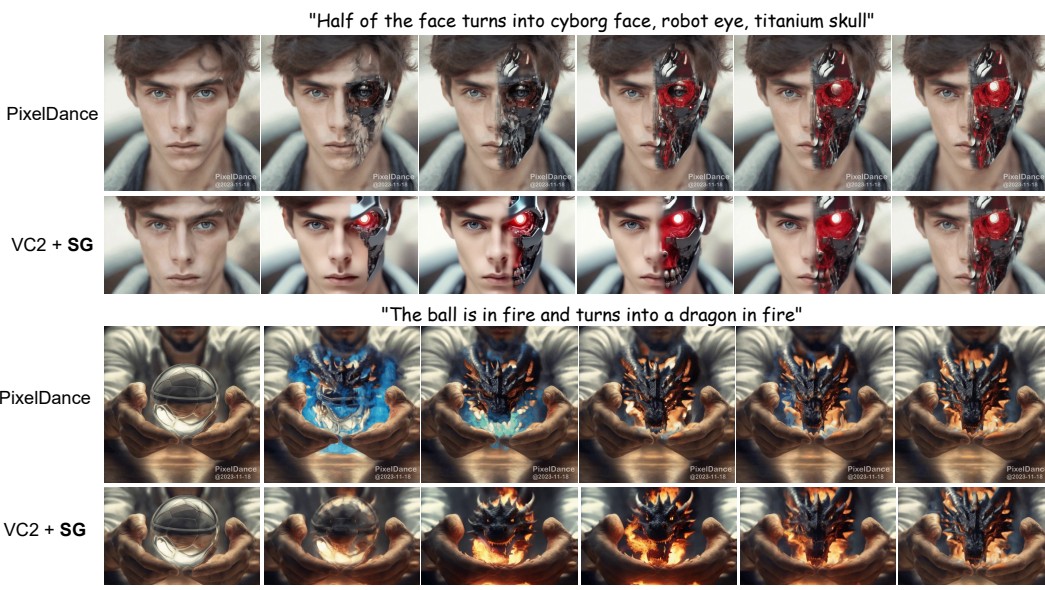

Figure 8: Video generation conditioned on same first and last frames (II2V) by PixelDance Zeng et al. (2023) and VideoCrafter2 Chen et al. (2024a) + State Guidance (VC2+SG). State Guidance allows to achieve competitive level of visual effects without training T2V model for II2V taks.

## A.6 BROADER IMPACT

The goal of our work is to tackle problem of prompt condition limitations in current video generation methods. State Guidance updates inference scheme of open source video generation models and pushes up the quality of their generated samples. Thus, any existing biases in these models, as long as potential harmful samples are explicitly inherited. Our method enhances quality of video generation, exhibiting a positive influence on video applications.

## A.7 Video & Prompt samples

We attached video samples according to quantative results in Table 3, Table 5, Table 6, and illustration figures and provide full description of our **Dynamic Scenes Benchmark**:

- **Videos from figures.** See videos_from_figures/ folder folder videos that were used for all illustrations in the paper. All subfolders contains video samples to each methods named accordingly to figure numbers;

- **Text-to-Video generation.** See example_videos_T2V/ folder. All subfolders contains video samples to each methods named accordingly to Table 3;

- **Text-to-Video generation for CogvideoX.** See example_videos_T2V/ CogvideoX. Each subfolder corresponds to a generation with the prompt specified in the folder name. Filenames denote following: *cogvideox_5b.mp4* - standard inference with short prompt, *cogvideox_5b_sg.mp4* - State Guidance inference with short prompts, *enhanced_cogvideox_5b.mp4* - standard inference with enhanced prompt, *enhanced_cogvideox_5b_sg.mp4* - State Guidance inference with enhanced prompts;

- **Image-to-Video generation.** See example_videos_I2V/ folder folder for II2V generation examples from Table 5;

- **Image-Image-to-Video generation.** See example_videos_II2V/ folder for II2V generation examples from Table 6);

- **Dynamic Scenes Benchmark.** We provide manual splitting of each prompt into initial state, end state, and transition state. See prompt labels and manually prompt splitting pages in Prompts_T2V.xlsx file;

- **Prompts to Image-to-Video generation.** See 111 prompts & initial frame paths for Image-to-Video generation quantitative results 5 in Prompts_I2V.xlsx file. In folder I2V_reference_frame we attached initial frames.

- **Prompts to Image-Image-to-Video generation.** See prompts that we use for Image-Image-to-Video generation quantitative results 5 in Prompts_II2V.xlsx file.

