# OpenReview forum: "State & Image Guidance: Teaching Old Text-to-Video Diffusion Models New Tricks"
_ICLR.cc/2025/Conference — Submitted to ICLR 2025_

### Official Review · Reviewer_NpS9 · 2024-10-29

**Soundness:** 2
**Presentation:** 2
**Contribution:** 2
**Rating:** 5
**Confidence:** 3

**Summary:**

The paper describes a method for Text To Video (T2V) generation.
The method proposes two extensions: State and Image guidance.
State Guidance uses state triplets (initial, current, last) to help T2V generate the proper video frame.
Image guidance injects noise in the early stages of the diffusion model to steer it in the right direction.
This is then used to: generate more dynamic video sequences, as well as zero shot video generation from a single image, and a video interpolating between two images.

**Strengths:**

The extensions are reasonable and the various experiments does show nice videos produced by the system.

In addition, a new dataset is introduced that, hopefully, will help future contributions to the field.

**Weaknesses:**

The paper feels rushed. (The caption of the teaser figure misplaced the text for sub-figures (B) and (C)).

I wonder what is the novelty of the proposed method given the "Make Pixels Dance" (CVPR'24) paper.
They, too, use a triplet state representation to encourage better video synthesis.
Yet, I could not find a direct comparison. Can the authors explain why?

There are many results and one must appreciate the work done by the authors, but it is extremely difficult to follow the experimental results and appreciate the contributions. For example,

1. Please add a reference to the different methods shown in the various tables.
2. I'm not sure the ablation experiments should appear in the main text.
3. The supplemental material is difficult to navigate. There are many folders and no easy way to navigate and compare the different results presented there.
4. Table 3: The boldface numbers are confusing as they only refer to the method without SG. Yet, in almost each column there is a better alternative, so it's difficult to judge the overall quality of the results.
5. Table 5: It is confusing to compare VC2+IG with three thresholds to TI2V-Zero and then highlight in bold different measures for different thresholds.

**Questions:**

Can the authors elaborate on the comparison to "Make Pixels Dance"?

---

> ### Author Response · Authors · 2024-11-14
>
> Dear Reviewer NpS9,
>
> Thank you for your thoughtful review and for recognizing the strengths of our work. We appreciate your specific feedback and are committed to addressing each of your concerns to improve the clarity and rigor of our paper.
>
> ### **Weakness 1**
>
> **Reviewer's Comment:** *"The paper feels rushed. (The caption of the teaser figure misplaced the text for sub-figures (B) and (C))."*
>
> **Response:** We apologize for the typo in the caption of the teaser figure, which misplaced the text for subfigures (B) and (C). We will correct this in the revised version. If there are other areas where the presentation could be improved, we welcome your specific suggestions to improve the clarity of our paper.
>
> ### **Weakness 2**
>
> **Reviewer's Comment:** *“What is the novelty of the proposed method given the "Make Pixels Dance" (CVPR'24) paper”*
>
> **Response:** We understand your concern about the novelty of our method compared to the paper "Make Pixels Dance" [1] (PixelDance). Although both papers use state triplets, approach is significantly different:
>
> - **PixelDance:** Utilizes ⟨first frame, text prompt, last frame⟩ triplets in a custom architecture **requiring training from scratch**, aimed at **enhancing T2V model training and improving video dynamics**.
> - **State & Image Guidance:** Employs a **training-free scheme** with novel inference mechanics for existing T2V models like VideoCrafter2 [2]:
>     - *State Guidance*: Conditions on ⟨first state prompt, transition state prompt, last state prompt⟩ for **dynamic video scene generation**.
>     - *Image Guidance*: Conditions on ⟨prompt, reference⟩ duplet for **Zero-Shot I2V generation**.
>     - *Combination of State Guidance and Image Guidance*: Conditions on ⟨first frame, text prompt, last frame⟩ triplet for **Zero-Shot II2V generation**.
>
> While our Zero-Shot II2V pipeline is comparable to the II2V regime of PixelDance, the motivations and implementations differ significantly. For a detailed comparison with *"Make Pixels Dance",* please refer to **Appendix A.5 (lines 892-896 and Figure 8)**.
>
> ### **Weakness 3**
>
> **Reviewer's Comment:** *“It is extremely difficult to follow the experimental results and appreciate the contributions.”*
>
> **Response:** We understand the importance of clarity in the presentation of experimental results and would like to address your concerns:
>
> 1. **References to different methods**: Due to space limitations in the tables, we have included references to the methods in the text preceding each table. Specifically:
>     - **Table 3**: The methods are listed on lines 395-396 and 403-405.
>     - **Table 5**: Methods are listed on lines 462-463.
>     - **Table 6**: Methods are listed on lines 484-485.
> 2. **Ablation studies in main text:** Including ablation studies in the main text is standard practice to highlight the impact of different components of the method. For example, recent papers such as "Ground-A-Video" [3] (ICLR'2024), "ControlVideo" [4] (ICLR'2024), and the aforementioned "Make Pixels Dance" [1] (CVPR'2024) include ablation studies in their experimental sections. We believe that this placement helps readers to better understand the papers.
> 3. **Supplementary Material Navigation**: Guidance for navigating the Supplementary Files is provided in **Appendix A.7**. We aim to make it as user-friendly as possible and will consider further improvements.
> 4. **The bold numbers in Table 3** highlight the performance improvements when using State Guidance (SG) over standard inference. Other methods are included for reference to show that even commercial models and models with multi-prompt inference may not perform as well.
> 5. **Table 5** shows the performance of our method with different "thresholds" (hyperparameter values) introduced in formula (6). We compare these variations with each other and with TI2V-Zero to show the flexibility and effectiveness of our approach (with the best results in bold).
>
> We hope these clarifications have addressed your concerns. We are committed to improving the presentation and clarity of our work and look forward to your response. Please don't hesitate to contact us if you have any further questions.
>
> Thank you again for your valuable feedback.
>
> ---
>
> [1] Zeng, Yan, et al. "Make Pixels Dance: High-Dynamic Video Generation." (CVPR’2024).
>
> [2] Chen H. et al. "VideoCrafter2: Overcoming Data Limitations for High-Quality Video Diffusion Models." (CVPR’2024).
>
> [3] Jeong H., Ye J. C. "Ground-a-video: Zero-shot grounded video editing using text-to-image diffusion models." (ICLR’2024).
>
> [4] Zhang Y. et al. "ControlVideo: Training-free Controllable Text-to-Video Generation." (ICLR’2024).

---

> > ### Author Response · Authors · 2024-11-25
> > **Gentle reminder - less than 2 days left for the author-reviewer discussion**
> >
> > Dear Reviewer NpS9,
> >
> > Thank you again for your constructive feedback and time/effort reviewing our paper. Since the rebuttal period is ending in less than two days, please let us know if our responses have addressed your concerns. We are happy to engage in further discussion to provide more clarifications if needed.
> >
> > Warm regards,
> >
> > The Authors

---

> > ### Comment · Reviewer_NpS9 · 2024-11-27
> >
> > Thanks for clarifying the difference with respect to PixelDance.
> > I have no further questions.

---

> > > ### Author Response · Authors · 2024-11-28
> > >
> > > Dear Reviewer NpS9,
> > >
> > > Thank you for accepting our clarification regarding the differences between our method and PixelDance.
> > >
> > > However, we noticed that our submission currently holds a Rating: 5, which is below the acceptance mark for this conference. Given that you have acknowledged our explanations and the revisions made, we kindly request a reevaluation of our rating. If there are remaining concerns, please provide a detailed justification for this assessment.
> > >
> > > Thank you for your consideration.
> > >
> > > Kind regards,
> > > The Authors

---

> > > > ### Author Response · Authors · 2024-12-02
> > > >
> > > > Dear Reviewer NpS9,
> > > >
> > > > We noticed that our previous message has not received a response. With less than 24 hours remaining in the discussion period, we kindly invite you to address our request for a reevaluation of our rating or provide further clarification on any remaining concerns.
> > > >
> > > > Thank you for your consideration.
> > > >
> > > > Kind regards,
> > > > The Authors

---

### Official Review · Reviewer_EnJp · 2024-11-04

**Soundness:** 3
**Presentation:** 2
**Contribution:** 2
**Rating:** 5
**Confidence:** 3

**Summary:**

This paper proposes a novel video generation guidance method capable of producing videos with drastically different cross-frame descriptions, such as texture changes, morph transformations, and large motions. The core innovation of the method is the state triplet, which decomposes the video into different phases, each with its own distinct description. The state triplet can be generated either from a large language model (LLM) or manually. The proposed method is training-free and has been evaluated on a new video benchmark, achieving performance comparable to methods that require task-specific training.

**Strengths:**

The proposed method is practically useful given that:

* It enables the transformations that would be very difficult to model with pretrained models, such as morph transformation, drastic texture changes over frames.

* The method is training-free and does not require text-video pairs with the target motion patterns, which is hard to collect in scale. The training free method achieves comparable performance as the compared training-based method.

**Weaknesses:**

- The T2V limitation mentioned in L202 is not convincing enough. The prompts used in the paper are relatively simple, consisting of only one or two sentences. Some related works, such as CogVideoX [1], utilize DiT-based structures and demonstrate that detailed prompts significantly improve video generation quality, both in appearance and motion. Therefore, if we consider a scenario with both detailed video captions and a DiT-based model that relies less on explicit per-frame modeling due to (1) temporal compression in the tokenizer, and (2) stronger spatial-temporal modeling capabilities (i.e., cross-frame modeling rather than per-frame), the limitation highlighted for the T2V model becomes less relevant. This is because we would not be restricted by a limited T2I model (point 2) and would benefit from enhanced spatial-temporal modeling (point 1).

- The paper lacks key information on how the transition order is maintained. While Eq. 1 models the joint conditional distribution given the prompt triplet, it does not specify how the generated images are constrained to follow the prompt order: initial -> transition -> final stage.  Ensuring this sequential alignment is crucial for achieving controllability and realism in the generated video.

- As mentioned in the limitation section in the supplementary, the method introduces additional hyper-parameters, such as the guidance scale at for the triplet states. Tweaking those hyper-parameter would be a case-specific effort and paper does not propose a principled approach for estimating/optimizing those hyper-parameters.

- As mentioned in the first item, there are strong models taking much more descriptive prompt as input for video generation. However, the paper does not include the comparison with those methods. The lack of this comparison makes the claim about the T2V limitation and the proposed method less convincing.

[1] CogVideoX: Text-to-Video Diffusion Models with An Expert Transformer. Yang, Zhuoyi and Teng, Jiayan and Zheng, Wendi and Ding, Ming and Huang, Shiyu and Xu, Jiazheng and Yang, Yuanming and Hong, Wenyi and Zhang, Xiaohan and Feng, Guanyu and others. arXiv preprint arXiv:2408.06072

**Questions:**

In addition to the key information missing mentioned above, I have several questions related to the details of the paper:

1. How to handle different combinations of text and image prompt? For example, what if we have the triplet description and only the end frame of the generated image? How is this case different from the case where we have the triplet description and only the first frame of the generated image?

2. How is the method able to handle the morph transform even although the pretrained model is rarely trained on videos with morphism since it is not common? More discussion on this would be appreciated.

3. Have the authors try a more detailed caption vs. the simple ones used in the paper? Will that lead to better motion?

---

> ### Author Response · Authors · 2024-11-15
>
> Dear Reviewer EnJp,
>
> Thank you for your valuable feedback. We would like to address some of your concerns and initiate a discussion.
>
> Regarding your points related to the CogVideoX paper (**Weakness 1**, **Weakness 4**, and **Question 3**), we would like to clarify that CogVideoX was released only 1.5 months before the submission deadline. According to general submission guidelines, it is acceptable not to consider such recent developments in our initial submission. Nevertheless, we are currently conducting detailed experiments with this model and will provide thorough responses within a week.
>
> Thank you for your understanding and patience. Responses to all other weaknesses and questions (**Weaknesses 2, 3** and **Questions 1, 2**) are presented below.
>
> ### **Weakness 2**
> **Reviewer's Comment:** *“The paper lacks key information on how the transition order is maintained…”*
>
> **Response:** The transition order is maintained by scheduling frame-wise hyperparameters $\gamma_{is}^f$ ,$\gamma_{ts}^f$, and $\gamma_{fs}^f$ as detailed in Eq. 2. These hyperparameters influence each frame's state:
>
> - $\gamma_{is}^f$ is highest at the first frame and decreases over time.
> - $\gamma_{ts}^f$ peaks in the middle and is lowest at the beginning and end.
> - $\gamma_{fs}^f$ is lowest at the first frame and increases towards the final frame.
> - We make sure that $\gamma_{is}^f +\gamma_{ts}^f+\gamma_{fs}^f=1$ for all frames $f$ to keep the data distribution consistent.
>
> This guidance schedule ensures the sequential alignment from the initial state, through the transition, to the final stage. We also utilize a Guidance Interval to enhance video consistency. For more details, please refer to Lines 226-252 and Table 1.
>
> ### **Weakness 3**
> **Reviewer's Comment:** *“As mentioned in the limitation section in the supplementary, the method introduces additional hyper-parameters...”*
>
> **Response:** Our method indeed introduces two new hyperparameters: *guidance schedule* and *guidance interval* $\xi$. These parameters significantly enhance control over video generation. The paper contains a detailed analysis of each hyperparameter and provides recommendations for their optimal values.
>
> *Guidance Schedule*:
>
> - **Dynamic T2V Task:** A variety of schedules were tested, and the Negative Linear schedule shows best scene dynamics compared to other methods (see Table 4).
> - **Zero-Shot II2V Task:** The Partial quadratic schedule ensures better consistency with reference images.
>
> *Guidance Interval* $\xi$:
>
> - **Dynamic T2V Task:** This parameter balances video dynamism and the connectedness of initial and final states. Our experiments show that $\xi=0.95$ yields optimal results (see Lines 412-419, Table 4).
> - **Zero-Shot I2V Task:** It balances image similarity and amount of motion, with $\xi \in [0.9, 0.98]$ being preferred (see Lines 457-460, Figure 4, Table 5).
> - **Zero-Shot II2V Task:** It ensures video quality and consistency, with $\xi=0.5$ being optimal based on our experiments (see Appendix A5, Lines 869-878, Table 8).
>
> While these hyperparameters add complexity, they are essential for ensuring higher control and quality in video generation.
>
> ### **Question 1**
>
> **Response:** As shown in Section 3.4, State Guidance (Eq. 3) with Image Guidance (Eq. 5) enables reference images as state descriptors. In our Zero-shot II2V pipeline, this applies when references are available for both start and end frames, as per Eq. 7. For scenarios with only one reference image and prompts for each state, we adjust Eq. 7 as follows:
>
> *Only end frame*:
>
> $\hat{\epsilon_\theta}^f(z_t,\langle p_{is},p_{ts}, \langle p_{fs},i_{fs}\rangle\rangle)=(w+1)\tilde{\epsilon_\theta}^f(z_t,\langle i_{is},p_{ts}, \langle p_{fs},i_{fs}\rangle\rangle)-w\epsilon_\theta^f(z_t, \varnothing)$
>
> $\tilde{\epsilon_\theta}^f(z_t,\langle p_{is},p_{ts},\langle p_{fs},i_{fs}\rangle\rangle)=\gamma_{is}^f\epsilon_\theta^f(z_t,p_{is})+\gamma_{ts}^f\epsilon_\theta^f(z_t, p_{ts})+\gamma_{fs}^f\bar{\epsilon_\theta}^f(z_t, p_{fs}, i_{fs})$
>
> *Only first frame*:
>
> $\hat{\epsilon_\theta}^f (z_t,\langle  \langle p_{is}, i_{is} \rangle, p_{ts}, p_{fs}\rangle)=(w+1)\tilde{\epsilon_\theta}^f(z_t,\langle \langle p_{is},i_{is}\rangle, p_{ts}, p_{fs}\rangle)-w\epsilon_\theta^f(z_t, \varnothing)$
>
> $\tilde{\epsilon_\theta}^f(z_t,\langle\langle p_{is}, i_{is}\rangle, p_{ts},p_{fs}\rangle)=\gamma_{is}^f\bar{\epsilon_\theta}^f(z_t, p_{is},i_{is})+\gamma_{ts}^f\epsilon_\theta^f(z_t,p_{ts})+\gamma_{fs}^f\epsilon_\theta^f(z_t,p_{fs})$
>
> ### **Question 2**
> **Response:** Our inference pipelines introduce an inductive bias that improves morph transform generation. We define a dynamic video scene as a trajectory with three states: *initial state*, *transition state*, and *final state*. The *initial* and *final states* are nearly static, resembling scenarios the T2V model is typically trained on. The guidance schedule ensures smooth transitions between these states, maintaining coherence and quality in morph transform.

---

> ### Author Response · Authors · 2024-11-22
>
> Dear Reviewer EnJp,
>
> We would like to supplement our previous response by addressing Weakness 1, Weakness 4, and Question 3. To provide a comprehensive answer, we conducted additional experiments using the CogVideoX-5B \[1\] model with both short and detailed prompts (all experiments are conducted with frozen random state, DDIM sampler with 50 steps, for both  with State Guidance we user Negative linear guidance schedule and $\xi = 0.95$). The detailed prompts were generated from the original short prompts and state prompts in our Dynamic T2V benchmark utilizing the official CogVideoX prompt enhancer available at: <https://huggingface.co/spaces/THUDM/CogVideoX-5B-Space>
>
>
>
> Table A. Quantitative evaluation of CogVideoX on Dynamic T2V scenes bench.
>
> | Model                          | Prompt type | TextSim  | OF Score  | TC        |
> |--------------------------------|-------------|----------|-----------|-----------|
> | CogVideoX (standard inference) | short       | 2.85     | **3.10** | **98.26** |
> | CogVideoX (SG inference)       | short       | **3.01** | 1.72      | 98.04     |
> | CogVideoX (standard inference) | enhanced    | 3.02     | **3.19** | **98.71** |
> | CogVideoX (SG inference)       | enhanced    | **3.16** | 2.19     | 98.32     |
>
>
>  In addition, we conducted a side-by-side user study comparing standard inference of CogVideoX and inference with State Guidance on short and detailed prompts. The results are summarized in Table B below:
>
> Table B.  Qualitative evaluation of CogVideoX on Dynamic T2V scenes bench.
>
> |Question| Prompt type | State Guidance inference, % | Equal, % | Standard inference, % |
> |----|------------------|-----------------------------|----------|-----------------------|
> | TA | short     | **42.8**                    | 22.6     | 34.6                  |
> | TA | enhanced | **57.2**                    | 17.9     | 24.9                  |
> | MQ | short    | 41.0                        | 11.1     | **47.9**              |
> | MQ | enhanced | **50.2**                    | 9.2      | 40.6                  |
>
> **Results analysis:**
>
> 1.  Our findings indicate that default CogVideoX-5B does encounter issues with generating dynamic video scenes on short prompts, which are elaborated on in our Weakness 1 response.
> 2. Table A confirms that utilizing enhanced prompts improves performance in Dynamic T2V generation.
> 3. Table A and Table B show that State Guidance improves dynamic video scene generation for both short and enhanced prompts.
>
>
> ### **Weakness 1**
>
> **Reviewer's Comment:** *“*The T2V limitation mentioned in L202 is not convincing enough…*”*
>
> **Response:** We conducted additional experiments with CogVideoX that reveal that:
>
> 1. *DiT-based Architectures*: While DiT-based T2V architectures like CogVideoX exhibit superior spatial-temporal modeling and flexible text conditioning, our experiments indicate that the default CogVideoX still struggles with generating dynamic video scenes (see Table A). This issue likely stems from pre-trained models not being extensively trained on dynamic scenes since it is not common in training data. However, we show that incorporating State Guidance (SG) significantly enhances CogVideoX’s capability in dynamic video scene generation (see Tables A and B).
>
> 2. *Detailed Prompts*: Our results (Table A) show that using detailed prompts improves Dynamic T2V generation quality. It's important to note that enhancing prompts is orthogonal to applying State Guidance. Our findings (Tables A and B) demonstrate that combining enhanced prompts with State Guidance further advances dynamic video scene generation.
>
> We hope these additional insights address your concerns effectively.
>
> ### **Weakness 4**
>
> **Reviewer's Comment:** *“*… there are strong models taking much more descriptive prompt as input for video generation…*”*
>
> **Response:** We have addressed this by providing additional experimental results in Table A and Table B above. For further details, please refer to the **Results Analysis** and the response to **Weakness 1**.
>
> ### **Question 3**
>
> **Reviewer's Comment:** *“*Have the authors try a more detailed caption vs. the simple ones used in the paper? Will that lead to better motion?*”*
>
> **Response:** Thank you for your question. In response, we conducted additional experiments and confirm that using more detailed prompts leads to better motion quality. For more details, please refer to the **Results Analysis** and the response to **Weakness 1**.
>
> ---
>
> [1] CogVideoX: Text-to-Video Diffusion Models with An Expert Transformer. Yang, Zhuoyi and Teng, Jiayan and Zheng, Wendi and Ding, Ming and Huang, Shiyu and Xu, Jiazheng and Yang, Yuanming and Hong, Wenyi and Zhang, Xiaohan and Feng, Guanyu and others. arXiv preprint arXiv:2408.06072

---

> > ### Author Response · Authors · 2024-11-25
> > **Gentle reminder - less than 2 days left for the author-reviewer discussion**
> >
> > Dear Reviewer EnJp,
> >
> > Thank you again for your constructive feedback and time/effort reviewing our paper. Since the rebuttal period is ending in less than two days, please let us know if our responses have addressed your concerns. We are happy to engage in further discussion to provide more clarifications if needed.
> >
> > Warm regards,
> >
> > The Authors

---

> > > ### Author Response · Authors · 2024-11-28
> > >
> > > Dear Reviewer EnJp,
> > >
> > >    We believe we have addressed your concerns, backed by additional experiments, and revised the manuscript. With 5 days remaining in the extended discussion period, we welcome any further discussion or clarification you might need.
> > >
> > >    Best regards,
> > >    The Authors

---

> > ### Comment · Reviewer_EnJp · 2024-11-29
> >
> > Thanks for providing additional experiments for the rebuttal. While I appreciate the efforts the author put on the additional experiment for addressing the concerns, I'm still not convinced:
> >
> > 1. "Our findings indicate that default CogVideoX-5B does encounter issues with generating dynamic video scenes on short prompts, which are elaborated on in our Weakness 1 response.":
> > One of the key observations from CogVideoX paper, like other SOTA models, is that we need detailed prompts for generating high quality videos. I'm not sure if it is fair to say that it is a limitation of such models given that extending the prompt is not difficult.
> >
> > 2. The additional experiments for the rebuttal on using more detailed prompts with SOTA models for addressing the concerns are appreciated. But given that they are important but missing during the submission, it is unfair to raise the score. In addition, what are the output from the prompts enhancer for the given short prompts? Do they reflect precisely the state transition? It requires lots of additional contents for the main paper to make the additional experiments for the more detailed prompts convincing.
> >
> > As a result, I will keep my original rating.

---

> > > ### Author Response · Authors · 2024-11-29
> > >
> > > Dear Reviewer EnJp,
> > >
> > >
> > > Thank you for your feedback and for taking the time to review our rebuttal. We appreciate your insights and would like to address the points you raised.
> > >
> > >
> > > 1. **Missing CogVideoX in the initial submission:**
> > >
> > > - According to ICLR policy, *"if a paper was published (i.e., at a peer-reviewed venue) on or after July 1, 2024, authors are not required to compare their own work to that paper"* (see https://iclr.cc/Conferences/2025/FAQ for details). The first version of the CogVideoX paper \[1\] was published on August 12, 2024, making it permissible to exclude it from our initial submission.
> > > - Recognizing the significance of your concern, we conducted extensive experiments with the CogVideoX architecture and prompt enhancement. These results are included in the rebuttal and the revised version of the paper available on OpenReview.
> > > - We believe it is unfair to lower the score for not including CogVideoX initially, given the policy and our subsequent efforts to address this.
> > >
> > >
> > > 2. **Detailed Prompts Clarification:**
> > >
> > > - Despite community observations suggesting that detailed prompts improve video generation quality, there is no quantitative analysis supporting this. The CogVideoX paper \[1\] uses detailed prompts to align training and test prompts, but does not focus on comparing short versus detailed prompts directly.
> > > - Our experiments show that while detailed prompts enhance visual appeal, they do not fully solve the challenge of generating dynamic scenes. Please refer to the "example_videos_T2V/CogvideoX" folder in the Supplementary Material for generation examples.
> > > - As previously stated, enhancing prompts is orthogonal to applying State Guidance. As detailed in Tables A and B (previous response), combining enhanced prompts with State Guidance improves dynamic video scene generation.
> > >
> > >
> > > 3. **Outputs of prompt enhancer:**
> > >
> > > - According to our observation, prompt enhancement adds more descriptive details but does not alter the core action within the scene. This demonstrates that while the prompts are more vivid and detailed, the fundamental actions remain unchanged.
> > >
> > >
> > >
> > > Considering our response, we kindly ask the reviewer to reconsider the evaluation of our paper. We have thoroughly addressed all concerns with substantial efforts, including additional experiments and evaluations. We are ready to provide any further clarifications if needed.
> > >
> > >
> > >
> > > Kind regards,
> > >
> > > The Authors
> > >
> > >
> > > [1] CogVideoX: Text-to-Video Diffusion Models with An Expert Transformer. Yang, Zhuoyi and Teng, Jiayan and Zheng, Wendi and Ding, Ming and Huang, Shiyu and Xu, Jiazheng and Yang, Yuanming and Hong, Wenyi and Zhang, Xiaohan and Feng, Guanyu and others. arXiv preprint arXiv:2408.06072

---

> > > > ### Author Response · Authors · 2024-12-02
> > > >
> > > > Dear Reviewer EnJp,
> > > >
> > > > We noticed that our previous message has not received a response. With the discussion period nearing its end, we kindly invite you to engage with us while there is still time.
> > > >
> > > > Kind regards,
> > > > The Authors

---

### Official Review · Reviewer_sBeJ · 2024-11-04

**Soundness:** 3
**Presentation:** 3
**Contribution:** 2
**Rating:** 8
**Confidence:** 5

**Summary:**

# Summary

The paper proposes a training-free framework for generating better motion dynamics and adding image conditions with existing pre-trained T2V models. Extensive experiments demonstrate the effectiveness of the proposed framework


EDIT:

**Strengths:**

# Strengths

- The proposed framework is training-free, and indeed achieves better motion dynamics for the mentioned types of prompts
- The proposed framework outperforms the mentioned baselines

**Weaknesses:**

# Weaknesses

- In Sec. 3, the definition of diffusion models might be incorrect
- The idea of state guidance seems similar to the deforum-like technique used in the stable diffusion user community
- While the guidance schedule seems reasonable, the paper does not mention how it was designed/selected
- Entries in the proposed dynamic scenes benchmark consist of three states tailored for the proposed framework, resulting in inconsistent experiments settings when compared with other baselines which does not support this type of inputs. In that case, is unclear how reliable the proposed benchmark is
- The paper does not mention how generated results were selected. Considering diffusion models can generate various of results from the same input conditions from different seeds, it would be better to report mean+std for each metric and report the success rate of each generation
- For II2V experiments, the paper does not compare with SEINE
- The scale of user study seems relatively limited and the design of user study seems flawed: "more changes" in question 2 does not necessarily indicate the result is better in terms of visual quality. The results with flickering artifacts and incorrect/unfavourable color changes could also be considered as "more changes"

# Other comments (not weaknesses)

- The paper coined a new term II2V, which is actually frame inbetweening/interpolation

**Questions:**

Please refer to the weaknesses section

---

> ### Author Response · Authors · 2024-11-14
>
> Dear Reviewer sBeJ,
>
> Thank you for your valuable feedback. We would like to offer a brief response to your concerns regarding our submission and to initiate a discussion. Note that the Weakness 4 and Weakness 6 responses will be further supplemented by the results of the experiments we are currently working on.
>
> ### **Weakness 1**
>
> **Reviewer's Comment:** *“*In Sec. 3, the definition of diffusion models might be incorrect*”*
>
> **Response:** Unfortunately, it is difficult to localize the incorrectness of the definition. Could you please specify this comment?
>
> ### **Weakness 2**
>
> **Reviewer's Comment:** *“*The idea of state guidance seems similar to the deforum-like technique used in the stable diffusion user community*”*
>
> **Response:** State Guidance differs from Deforum in the following ways:
>
> 1. Models and Objectives:
>
> - *Deforum*: Uses Image Diffusion models to animate single static images.
> - *State Guidance*: Utilizes Video Diffusion models specifically designed for generating dynamic video scenes.
>
> 2. Generation Process:
>
> - *Deforum*: Iteratively transforms frames by adding and removing noise with Text-to-Image models.
> - *State Guidance*: Applies a concept guidance strength schedule over time to ensure smooth transitions between initial, transition, and final states.
>
> Additionally, our method incorporates Image Guidance and combines it with State Guidance to enhance video quality and consistency (see Lines 100-107).
>
> ### **Weakness 3**
>
> **Reviewer's Comment:** *“*Entries in the proposed dynamic scenes benchmark consist of three states tailored for the proposed framework…*”*
>
> **Response:** Our Dynamic Scenes Benchmark comprises 106 **single prompts** describing video scenes with significant changes, compatible with all standard Text-to-Video (T2V) models. The state prompt triplets are specifically designed for our State Guidance approach, utilizing pre-trained models and requiring structured input.
>
> Regarding experimental consistency, as shown in Table 3, we evaluated standard pre-trained T2V models using their typical inference method and our State Guidance approach without altering their architectures or weights. State Guidance enables models to efficiently incorporate state prompt triplets, significantly enhancing their ability to generate dynamic video scenes.
>
> Additionally, Table 3 includes results from two closed-source commercial T2V models and two models that use multiple prompts from large language models (LLMs), which also struggle with dynamic scene generation, confirming the reliability and relevance of our benchmark.
>
> ### **Weakness 4**
>
> **Reviewer's Comment:** *“*The paper does not mention how generated results were selected…*”*
>
> **Response:**  We generated all videos using a fixed random seed to ensure consistency, and preliminary tests showed negligible metric variation with different seeds. While reporting mean and standard deviation is ideal, the computational demands of diffusion-based video generation make it challenging and it is uncommon in the current literature.
>
> To address your concern, we will include the mean, standard deviation, and success rates for key results in Table 3 for both LaVie and VideoCrafter2 with and without State Guidance. We hope this addition meets your expectations.
>
> ### **Weakness 5**
>
> **Reviewer's Comment:** *“*For II2V experiments, the paper does not compare with SEINE*”*
>
> **Response:** Due to page limitations, we were initially unable to include this comparison. We recognize its importance and will incorporate it in the revised manuscript. Below is the comparison, where M stands for Metamorphosis, A for Animation, and O for Overall performance.
>
> | Method | FID (M) | PPL (M) | FID (A) | PPL (A) | FID (O) | PPL (O) |
> | --- | --- | --- | --- | --- | --- | --- |
> | SEINE | 82.03 | 47.72 | 48.25 | 16.26 | 67.60 | 39.33 |
> | S&IG (Ours) | **35.46** | **12.26** | **31.44** | **6.58** | **30.15** | **10.75** |
>
> ### **Weakness 6**
>
> **Reviewer's Comment:** *“*The scale of user study seems relatively limited and the design of user study seems flawed*”*
>
> **Response:** Thank you for your feedback. We are revising the study to improve its robustness and will include the updated design in our revised submission next week.
>
> ### **Comment 1**
>
> **Reviewer's Comment:** *“*The paper coined a new term II2V, which is actually frame inbetweening/interpolation*”*
>
> **Response:** We define II2V as generating transitions between two input reference frames (see Lines 49-50). This term aligns with related concepts like I2V and Dynamic T2V, ensuring consistency and clarity within our framework.
>
> ----
> Please let us know if our answers meet your expectations or if there are other areas that need clarification. We look forward to your continued feedback. Thank you again and we hope to hear from you soon.

---

> > ### Comment · Reviewer_sBeJ · 2024-11-21
> >
> > - W2/W5: I am satisfied with the responses provided, and these weaknesses are resolved
> >
> > ---
> >
> > - W1: If `A belongs to B`, can we formally state that `B is A`?
> >
> > - W3:  There still seems to be no guarantee that the inputs are consistent, especially in the learned semantic space. I understand that ensuring consistency might be challenging, given that the input formats are different. What happens if the three states are concatenated with temporal adverbials or conjunctions?
> >
> > ---
> >
> > - C1: Assigning a new name to a well-recognized concept still feels a bit unconventional. However, this is your choice as authors, and I do not consider it a weakness of the paper

---

> ### Author Response · Authors · 2024-11-22
>
> Dear Reviewer sBeJ,
>
> Thank you for providing response to our brief response and accepting our responses to W2/W5. We would like to supplement our brief response with detailed responses to W1, W3, W4, and W6.
>
> ### **Weakness 1**
>
> **Reviewer's Comment:** *“In Sec. 3, the definition of diffusion models might be incorrect…”*
>
> **Response:** If we understand you correctly, your comment is related to the connection between the Noise Conditional Score Network (NCSN [2]) and the Diffusion Model (DDPM [3]). The training goals of both methods can be mathematically linked. In particular, the denoising goal in DDPMs can be interpreted as a form of score matching under certain conditions.
>
> Section 2.2 of [1] states that the NCSN [2] and DDPM [3] ($L_{simple}$) goals are equivariant and both are a weighted sum of denoising score matching goals. Therefore, ***the NCSN objective simplifies to the DDPM objective***. Both objectives minimize the expected squared error between the true noise $\boldsymbol{\epsilon}$ and the model's noise prediction $\boldsymbol{\epsilon}_\theta(\mathbf{x}_t, t)$, so they're mathematically equivalent.
>
> ---
>
> [1] Song Y. et al. “Score-based generative modeling through stochastic differential equations” (ICLR’2021)
>
> [2] Song Y., Ermon S. “Generative modeling by estimating gradients of the data distribution (NeurIPS 2019)
>
> [3] Ho J., Jain A., Abbeel P. “Denoising diffusion probabilistic models” (NeurIPS 2020)
>
> ### **Weakness 3**
>
> **Reviewer's Comment:** *“*… What happens if the three states are concatenated with temporal adverbials or conjunctions?*”*
>
> **Response:**  We would like to clarify that the prompt triplet model does not introduce additional information beyond the original prompt. Instead, it explicitly separates the initial and final states from the original prompt, with the transition prompt either duplicating the original prompt or converting it into an action state. For instance, the original prompt *"Empty glass fills with water"* is transformed into the triplet: *<"An empty glass", "A glass is filling with water", "A glass with water">*.
>
> For experimental rigor, we tested VideoCrafter2 with triplet prompts concatenated using " , then " (results in the table below). This performed worse than using the original prompt directly:
>
> | Model | TextSim | OF Score | TC |
> | --- | --- | --- | --- |
> | VideoCrafter2 (original prompt) | 2.87 | 2.07 | 98.40 |
> | VideoCrafter2 (concat triplet) | 2.75 | 1.79 | **98.56** |
> | VideoCrafter2 (SG inference) | **3.18** | **3.83** | 97.40 |
>
> ### **Weakness 4**
>
> **Reviewer's Comment:** “The paper does not mention how generated results were selected...”
>
> **Response:**
>
> 1. Selection of generated results: All videos were generated with a fixed random seed for consistency. Preliminary tests showed minimal metric variation with different seeds.
> 2. Reporting mean+std: We evaluated the metrics from Table 3 for both LaVie and VideoCrafter2, with and without State Guidance, using five different random seeds. The results show low standard deviations, indicating robustness:
>
> | Model | TextSim | OF Score | TC |
> | --- | --- | --- | --- |
> | LaVie (standard inference) | 2.77 ± 0.04 | 4.77 ± 0.52 | **97.90 ± 0.20** |
> | LaVie (SG inference) | **3.14 ± 0.05** | **9.00 ± 0.72** | 96.40 ± 0.10 |
> | VideoCrafter2 (standard inference) | 2.84 ± 0.09 | 1.97 ± 0.10 | **98.42 ± 0.04** |
> | VideoCrafter2 (SG inference) | **3.12 ± 0.05** | **3.70 ± 0.25** | 97.30 ± 0.20 |
>
> The non-overlapping value intervals further affirm the consistency of our results.
>
> 3. Reporting success rate of each generation: Unfortunately, computing the success rate of each generation requires user involvement. However, our quantitative TextSim and qualitative TA metrics provide equivalent information.
>
> ### **Weakness 6**
>
> **Reviewer's Comment:** *“The scale of user study seems relatively limited and the design of user study seems flawed…”*
>
> **Response:** We have improved our user study methodology and conducted it on a larger set of users. Additionally, we have added the option "Equal" to each question to account for instances where users are unable to prefer one option over the other.
>
> Updated Instructions:
>
> - Question 1 (TA): "Which video better reflects the actions described in the text description?"
> - Question 2 (MQ): "Which video is more dynamic (has more action and events, including simultaneous events)?"
>
> For each side-by-side comparison, between 50 and 67 users participated, with each pair of videos assessed by at least 5 unique users. Below are the updated user study results:
>
> | Question | Model | State Guidance inference, % | Equal, % | Standard inference, % |
> | --- | --- | --- | --- | --- |
> | TA | LaVie | **70.6** | 13.0 | 16.4 |
> | TA | VideoCrafter2 | **66.7** | 22.2 | 11.1 |
> | MQ | LaVie | **74.3** | 5.1 | 20.6 |
> | MQ | VideoCrafter2 | **68.1** | 14.3 | 17.6 |

---

> > ### Author Response · Authors · 2024-11-25
> > **Gentle reminder - less than 2 days left for the author-reviewer discussion**
> >
> > Dear Reviewer sBeJ,
> >
> > Thank you again for your constructive feedback and time/effort reviewing our paper. Since the rebuttal period is ending in less than two days, please let us know if our responses have addressed your concerns. We are happy to engage in further discussion to provide more clarifications if needed.
> >
> > Warm regards,
> >
> > The Authors

---

> > > ### Author Response · Authors · 2024-11-28
> > >
> > > Dear Reviewer sBeJ,
> > >
> > > We apologize for inadvertently missing one of your concerns.
> > >
> > > ### **Weakness 7**
> > >
> > > Reviewer's Comment: "While the guidance schedule seems reasonable, the paper does not mention how it was designed/selected."
> > >
> > > Response: The guidance schedule is designed based on the following principles:
> > >
> > > - $\gamma_{is}^f$ starts highest at the first frame and decreases throughout the video.
> > > - $\gamma_{ts}^f$ has the lowest values at the beginning and end, peaking in the middle.
> > > - $\gamma{_fs}^f$ starts low and increases towards the final frame.
> > > - The sum $\gamma_{is}^f + \gamma_{ts}^f + \gamma_{fs}^f = 1$ for all frames $f$, ensuring consistency in data distribution.
> > >
> > > As shown in Table 1, $\gamma_{is}^f$ and $\gamma_{fs}^f$ are symmetric around the middle of the video. The schedules differ mainly in the functions defining $\gamma_{is}^f$ and $\gamma_{fs}^f$. We observed that a Negative Linear schedule works best for Dynamic T2V, enhancing dynamics and expressiveness. For Zero-shot-II2V, a Partial Quadratic schedule better aligns with conditional images.
> > >
> > > We also use a Guidance Interval to further improve video consistency. For more details, please refer to Lines 256-272 and Table 1 in the main text.
> > >
> > > Kind regards,
> > >
> > > The Authors

---

> > > > ### Author Response · Authors · 2024-11-28
> > > > **Summing up**
> > > >
> > > > Dear Reviewer sBeJ,
> > > >
> > > >
> > > > We believe we have addressed all your concerns and revised our submission accordingly. With 5 days remaining in the extended discussion period, we welcome any further discussion or clarification you might need.
> > > >
> > > >
> > > >
> > > >
> > > >
> > > >
> > > > Best regards,
> > > >
> > > > The Authors

---

> > > > > ### Author Response · Authors · 2024-12-02
> > > > >
> > > > > Dear Reviewer sBeJ,
> > > > >
> > > > > With less than 24 hours remaining in the author-reviewer discussion period, could you please confirm if all your concerns have been fully addressed? If you have any additional questions or need further clarification, please let us know.
> > > > >
> > > > > Best regards,
> > > > > The Authors

---

> > > > ### Comment · Reviewer_sBeJ · 2024-12-03
> > > >
> > > > W3, W4, W7: I am satisfied with the responses provided, and these weaknesses have been resolved
> > > > Specifically for W3, your results align with the observation that CLIP text embeddings are biased towards visual context rather than motion context. The proposed framework indeed provides a workaround for this bias
> > > >
> > > > Based on your current responses and the feedback from other reviewers, I will raise my rating to above 5
> > > >
> > > > ------
> > > >
> > > > W6: The major issue remains that your user study design does not necessarily indicate that the chosen results have better visual quality. Additionally, the use of "equal" introduces ambiguity. A two-alternative forced-choice design might be a better choice.
> > > >
> > > > Comments for W1: Please also review L189-192

---

> > > > > ### Author Response · Authors · 2024-12-03
> > > > >
> > > > > Dear Reviewer sBeJ,
> > > > >
> > > > > Thank you for your thoughtful review and constructive feedback. We appreciate your commitment and would like to address your comments on Weakness 1 and Weakness 6.
> > > > >
> > > > > ### **Weakness 1**
> > > > >
> > > > > We appreciate your insight regarding this section. We have made improvements to the Background section and will to include them in the revised version of the paper to enhance clarity:
> > > > >
> > > > > *"Diffusion Models perform image generation by iteratively denoising data corrupted with Gaussian noise. A sample image $x_0$ is progressively corrupted according to a predefined Markov chain: $q(x_t|x_{t-1}) = \mathcal{N}(x_t; \sqrt{1-\beta_t}x_{t-1}, \beta_tI),$ where $\beta_t$ controls the noise level at each step. The forward diffusion process can be expressed in closed form as $ x_t = \sqrt{\alpha_t}x_0 + \sqrt{1-\alpha_t}\epsilon $, where $\epsilon \sim \mathcal{N}(0, \mathbf{I})$ and $\alpha_t = \prod_{i=1}^t (1 - \beta_t)$. The reverse diffusion process, parameterized by a neural network $\epsilon_\theta$, aims to recover $x_0$ by iteratively denoising $x_t$ from pure noise and predicting the noise $\epsilon$. It is learned by minimizing the mean squared error loss: $ \mathcal{L} = \mathbb{E}_{x_0, c, \epsilon, t}[ | \epsilon - \epsilon_0 (x_t, t, c) |_2^2],$ conditioned on a text prompt $c$."*
> > > > >
> > > > > ### **Weakness 6**
> > > > >
> > > > > **User Study Questions:** Our user study is designed to validate our findings that State Guidance improves text alignment and video dynamics on the Dynamic Scene Benchmark. We focus on two specific questions to assess these aspects. Since visual quality is inherently tied to the generation model, which remains constant across options, we have opted not to include a specific visual quality question.
> > > > >
> > > > > **Inclusion of "Equal" Option:** The "equal" option allows participants to indicate uncertainty when choosing between two alternatives. We believe this is beneficial, as it minimizes random selection in cases where users are unsure, leading to more reliable data.
> > > > >
> > > > > We sincerely thank you for your commitment to raise your rating above 5 and look forward to any further updates.
> > > > >
> > > > > Warm regards,
> > > > >
> > > > > The Authors

---

### Author Response · Authors · 2024-11-22

We sincerely thank the reviewers for their constructive feedback and the time and effort spent reviewing our paper. In this rebuttal, we aim to provide clear and evidence-based responses to each reviewer's concerns. Please feel free to request further clarifications during the rebuttal period if needed.

---

### Author Response · Authors · 2024-11-28
**Manuscript Revision details**

Dear Reviewers,

Thank you for your insightful comments and suggestions on our manuscript. We have made the following revisions, highlighted in blue in the updated manuscript to improve clarity and address your feedback:

Reviewer sBeJ:
1. Added details on result selection in Section 4 (lines 365-367). Mean and standard deviation results for LaVie and VideoCrafter2 are now reported in Appendix A.3 (lines 863-865, Table 8).
2. Included a comparison with SEINE in the II2V experiment (Table 6).
3. Revised the user study methodology and conducted experiments with a larger user group (Section 4, Table 4, and Appendix A.3, lines 855-860).

Reviewer EnJp:
1. Updated T2V limitations to discuss the scarcity of dynamic video scenes in training datasets (Section 1, lines 060-072; Section 3, lines 200-220).
2. Added experiments with the DiT-based model (CogVideoX) and detailed prompts, incorporating prompt enhancement from the CogVideoX web UI (Section 4, lines 405-413; Tables 3 and 4; and Appendix A.7, lines 1036-1041).

Reviewer NpS9:
1. Fixed misplaced labels for sub-figures (B) and (C) in the teaser figure.
2. Added references to the various methods in multiple tables (Tables 3, 4, 5, 6, 8).
3. Moved the table evaluating different guidance schedules and $\xi$ parameter values to Appendix A.3 (Table 7, lines 830-838).

Best regards,

The Authors

---

### Author Response · Authors · 2024-12-03
**Author-Reviewer Discussion Summary**

Dear Reviewers and Area Chairs,

As the discussion period concludes, we would like to summarize our interactions with the reviewers.

We extend our gratitude to reviewers sBeJ, EnJp, and NpS9 for their time and feedback on our submission. All three initially rated our paper a 5—marginally below the acceptance threshold. However, they positively recognized the proposed extensions to the diffusion-based video generation model, noting their effectiveness across various tasks, the high quality of the generated videos, the training-free nature of our method, and the proposed benchmark.

During the discussion period, we did our best to  address all reviewers' concerns:

### **Reviewer sBeJ:**

1. Refined the definition of the diffusion model.
2. Clarified the differences between our method and Deforum.
3. Provided additional baseline comparisons to ensure experimental fairness.
4. Explained the selection of generated results and conducted further experiments to demonstrate statistical consistency.
5. Added comparisons with SEINE in the II2V experiment.
6. Improved the user study methodology and expanded the participant pool.
7. Explained the design of the guidance schedule.

**Result:** Reviewer sBeJ has committed to increasing their rating.

### **Reviewer EnJp:**

1. Conducted additional experiments with CogVideoX on short and enhanced prompts.
2. Clarified how transition order is maintained in State Guidance.
3. Provided recommended values for new hyperparameters.
4. Explained our method's application in new scenarios and the surprising effectiveness of the morph transform.

**Result:** Reviewer EnJp decided to maintain their original rating.

### **Reviewer NpS9:**

1. We clarified aspects of the paper text and experiments.
2. Explained the differences between our method and "Make Pixels Dance."

**Result:** Reviewer NpS9 accepted our clarifications and indicated they had no further questions but did not provide additional information regarding their rating.

Thank you for your attention.

Best regards,

The Authors

---

### Meta-Review · Area_Chair_PeGa · 2024-12-21

**Metareview:**

The paper introduces two sampling methods with state and image guidance to enhance the generative capabilities of text-to-video diffusion models. The authors demonstrate video generation scenarios such as generating dynamic video scenes, zero-shot generation conditioned on the input image, and zero-shot generation conditioned on the first and last frames.

The paper initially received ratings of 5, 5, and 5. After the rebuttal, one of the reviewers increased the rating to 8, and the final ratings became 8, 5, and 5. During the rebuttal phase, the authors presented additional experimental results to address the reviewers' concerns. However, the other reviewers maintained their ratings despite having no further questions. The final ratings showed that both reviewers were not persuaded by the authors' responses. The area chair valued the expertise of these two experienced reviewers and found no strong reason to disagree with their recommendations. Consequently, the area chair regretfully recommended rejecting the paper.

**Additional Comments On Reviewer Discussion:**

* The concern about the definition of diffusion models raised by **Reviewer sBeJ** was not updated in the revised manuscript.

* **Reviewer EnJp** was not sure if it is fair to say that the need for detailed prompts is a limitation of SOTA models, given that extending the prompt is not difficult. It should be noted that not including CogVideoX initially was not considered a reason when making the final decision. **Reviewer EnJp** simply referred to CogVideoX as an example of the SOTA models that could benefit from detailed prompts for a high-quality generation. Therefore, the concern raised by **Reviewer EnJp** is still relevant.
> It requires lots of additional contents for the main paper to make the additional experiments for the more detailed prompts convincing.

* As pointed out by **Reviewer NpS9**, failing to refer to and compare with "Make Pixels Dance" (CVPR'24) paper during the submission could pose an issue.

---

> ### Public Comment · ~Konstantin_Sobolev2 · 2025-02-25
> **A comment on the review process**
>
> Dear ICLR Review Committee,
>
> As the author of this paper, I would like to express some concerns with the review process and the quality of the review comments.
>
> Initially, the paper received ratings of 5, 5, and 5. After extensive discussion with Reviewer sBeJ and conducting additional experiments, Reviewer sBeJ appreciated the significance of our work and increased their rating to 8. It is important to note that this reviewer had the highest confidence rating of 5 among all reviewers. However, it appears that the Area Chair (AC) based the final rating predominantly on the opinions of Reviewer EnJp and Reviewer NpS9, who showed limited engagement during the review process and had lower confidence ratings of 3.
>
> I would also like to highlight three specific points related to reviewers' comments during the discussion:
>
> 1. Definition of Diffusion Models:
>    - AC noted a concern regarding the definition of diffusion models which was not updated in the revised manuscript. We wrote about this on December 3; however, at that point, the submission editing window was closed. Therefore, we were talking about updating the manuscript after acceptance.
>
>
> 2. Detailed Prompts and SOTA Models:
>    - Reviewer EnJp expressed uncertainty about the limitations of SOTA models needing detailed prompts, arguing that extending the prompt is not difficult. It is important to note that by July 1, 2024, no open-source DiT for video generation used an alternative text conditioning mechanism to frame-wise cross attention and detailed prompts. Additionally, Reviewer EnJp did not provide other models, other than CogVideoX, to support their claim, rendering this concern less applicable.
>
> 3. Comparison with "Make Pixels Dance" (CVPR'24):
>    - Reviewer NpS9 raised an issue about not referring to and comparing with the "Make Pixels Dance" paper. This comparison was, in fact, included in the original submission. During the rebuttal, we clarified this and directed the reviewer's attention to the relevant section in the appendix, providing additional explanations about the differences between the methods.

---

### Decision · Program_Chairs · 2025-01-22

Reject